# SELECTING TREATMENT EFFECTS MODELS FOR DOMAIN ADAPTATION USING CAUSAL KNOWLEDGE

## ABSTRACT

Selecting causal inference models for estimating individualized treatment effects (ITE) from observational data presents a unique challenge since the counterfactual outcomes are never observed. The problem is challenged further in the unsupervised domain adaptation (UDA) setting where we only have access to labeled samples in the source domain, but desire selecting a model that achieves good performance on a target domain for which only unlabeled samples are available. Existing techniques for UDA model selection are designed for the predictive setting. These methods examine discriminative density ratios between the input covariates in the source and target domain and do not factor in the model's predictions in the target domain. Because of this, two models with identical performance on the source domain would receive the same risk score by existing methods, but in reality, have significantly different performance on the test domain. We leverage the invariance of causal structures across domains to introduce a novel model selection metric specifically designed for ITE models under the UDA setting. In particular, we propose selecting models whose predictions of the effects of interventions satisfy known causal structures in the target domain. Experimentally, our method selects ITE models that are more robust to covariate shifts on several synthetic and real healthcare datasets, including on estimating the effect of ventilation in COVID-19 patients from different geographic locations.

## 1 INTRODUCTION

Causal inference models for estimating individualized treatment effects (ITE) are designed to provide actionable intelligence as part of decision support systems and, when deployed on mission-critical domains, such as healthcare, require safety and robustness above all (Shalit et al., 2017; Alaa & van der Schaar, 2017). In healthcare, it is often the case that the observational data used to train an ITE model may come from a setting where the distribution of patient features is different from the one in the deployment (target) environment, for example, when transferring models across hospitals or countries. Because of this, it is imperative to select ITE models that are robust to these covariate shifts across disparate patient populations. In this paper, we address the problem of *ITE model selection in the unsupervised domain adaptation (UDA)* setting where we have access to the response to treatments for patients on a source domain, and we desire to select ITE models that can reliably estimate treatment effects on a target domain containing only unlabeled data, i.e., patient features.

UDA has been successfully studied in the predictive setting to transfer knowledge from existing labeled data in the source domain to unlabeled target data (Ganin et al., 2016; Tzeng et al., 2017). In this context, several model selection scores have been proposed to select predictive models that are most robust to the covariate shifts between domains (Sugiyama et al., 2007; You et al., 2019). These methods approximate the performance of a model on the target domain (*target risk*) by weighting the performance on the validation set (*source risk*) with known (or estimated) density ratios.

However, ITE model selection for UDA differs significantly in comparison to selecting predictive models for UDA (Stuart et al., 2013). Notably, we can only approximate the estimated counterfactual error (Alaa & van der Schaar, 2019), since we only observe the factual outcome for the received treatment and cannot observe the counterfactual outcomes under other treatment options (Spirtes et al., 2000). Consequently, existing methods for selecting predictive models for UDA that compute a weighted sum of the validation error as a proxy of the target risk (You et al., 2019) is suboptimal for

selecting ITE models, as their validation error in itself is only an approximation of the model's ability to estimate counterfactual outcomes on the source domain.

To better approximate target risk, we propose to leverage the invariance of causal graphs across domains and select ITE models whose predictions of the treatment effects also satisfy known or discovered causal relationships. It is well-known that causality is a property of the physical world, and therefore the physical (functional) relationships between variables remain invariant across domains (Schoelkopf et al., 2012; Bareinboim & Pearl, 2016; Rojas-Carulla et al., 2018; Magliacane et al., 2018). As shown in Figure 1, we assume the existence of an underlying causal graph that describes the generating process of the observational data. We represent the selection bias present in the source observational datasets by arrows between the features $\{X_1, X_2\}$, and treatment $T$. In the target domain, we only have access

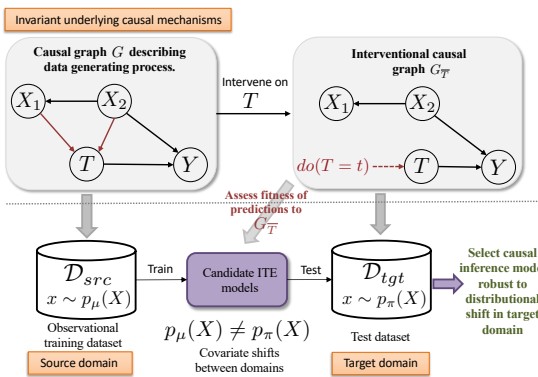

Figure 1: Method overview. We propose selecting ITE model whose predictions of the treatment effects on the target domain satisfy the causal relationships in the interventional causal graph $G_{\overline{T}}$.

to the patient features, and we want to estimate the patient outcome ($Y$) under different settings of the treatment (intervention). When performing such interventions, the causal structure remains unchanged except for the arrows into the treatment node, which are removed.

■ **Contributions.** To the best of our knowledge, we present the first UDA selection method specifically tailored for machine learning models that estimate ITE. Our ITE model selection score uniquely leverages the estimated patient outcomes under different treatment settings on the target domain by incorporating a measurement of how well these outcomes satisfy the causal relationships in the interventional causal graph $G_{\overline{T}}$. This measure, which we refer to as causal risk, is computed using a log-likelihood function quantifying the model predictions' fitness to the underlying causal graph. We provide a theoretical justification for using the causal risk, and we show that our proposed ITE model selection metric for UDA prefers models whose predictions satisfy the conditional independence relationships in $G_{\overline{T}}$ and are thus more robust to changes in the distribution of the patient features. We also show experimentally that adding the causal risk to existing state-of-the-art model selection scores for UDA results in selecting ITE models with improved performance on the target domain. We provide an illustrative example of model selection for several real-world datasets for UDA, including ventilator assignment for COVID-19.

## 2 RELATED WORKS

Our work is related to causal inference and domain adaptation. In this section, we describe existing methods for ITE estimation, UDA model selection in the predictive setting, and domain adaptation from a causal perspective.

■ **ITE models.** Recently, a large number of machine learning methods for estimating heterogeneous ITE from observational data have been developed, leveraging ideas from representation learning (Johansson et al., 2016; Shalit et al., 2017; Yao et al., 2018), adversarial training, (Yoon et al., 2018), causal random forests (Wager & Athey, 2018) and Gaussian processes (Alaa & van der Schaar, 2017; 2018). Nevertheless, no single model will achieve the best performance on all types of observational data (Dorie et al., 2019) and even for the same model, different hyperparameter settings or training iterations will yield different performance.

■ **ITE model selection.** Evaluating ITE models' performance is challenging since counterfactual data is unavailable, and consequently, the true causal effects cannot be computed. Several heuristics for estimating model performance have been used in practice (Schuler et al., 2018; Van der Laan & Robins, 2003). Factual model selection only computes the error of the ITE model in estimating the factual patient outcomes. Alternatively, inverse propensity weighted (IPTW) selection uses the estimated propensity score to weigh each sample's factual error and thus obtain an unbiased estimate (Van der Laan & Robins, 2003). Alaa & van der Schaar (2017) propose using influence functions to approximate ITE models' error in predicting both factual and counterfactual outcomes. Influence

function (IF) based validation currently represents the state-of-the-art method in selecting ITE models. However, existing ITE selection methods are not designed to select models robust to distributional changes in the patient populations, i.e., for domain adaptation.

■ **UDA model selection.** UDA is a special case of domain adaptation, where we have access to unlabeled samples from the test or target domain. Several methods for selecting predictive models for UDA have been proposed (Pan & Yang, 2010). Here we focus on the ones that can be adapted for the ITE setting. The first unsupervised model selection method was proposed by Long et al. (2018), who used Importance-Weighted Cross-Validation (IWCV) (Sugiyama et al., 2007) to select hyperparameters and models for covariate shift. IWCV requires that the importance weights (or density ratio) be provided or known ahead of time, which is not always feasible in practice. Later, Deep Embedded Validation (DEV), proposed by You et al. (2019), was built on IWCV by using a discriminative neural network to learn the target distribution density ratio to provide an unbiased estimation of the target risk with bounded variance. However, these proposed methods do not consider model predictions on the target domain and are agnostic of causal structure.

■ **Causal structure for domain adaptation.** Recently, Kyono & van der Schaar (2019) proposed Causal Assurance (CA) as a domain adaptation selection method for predictive models that leverages prior knowledge in the form of a causal graph. Because their work is centered around predictive models, it is suboptimal for ITE models, where the edges into the treatment (or intervention) will capture the selection bias of the observational data. Furthermore, their method does not allow for examining the target domain predictions, which is a key novelty of this work. We leverage $do$-calculus (Pearl, 2009) to manipulate the underlying directed acyclical graph (DAG) into an interventional DAG that more appropriately fits the ITE regime. More recently, researchers have focused on leveraging the causal structure for predictive models by identifying subsets of variables that serve as invariant conditionals (Rojas-Carulla et al., 2018; Magliacane et al., 2018).

## 3 PRELIMINARIES

### 3.1 INDIVIDUALIZED TREATMENT EFFECTS AND MODEL SELECTION FOR UDA

Consider a training dataset $\mathcal{D}_{src} = \{(x_i^{src}, t_i^{src}, y_i^{src})\}_{i=1}^{N_{src}}$ consisting of $N_{src}$ independent realizations, one for each individual $i$, of the random variables $(X, T, Y)$ drawn from the source joint distribution $p_\mu(X, T, Y)$. Let $p_\mu(X)$ be the marginal distribution of $X$. Assume that we also have access to a test dataset $\mathcal{D}_{tgt} = \{x_i^{tgt}\}_{i=1}^{N_{tgt}}$ from the target domain, consisting of $N_{tgt}$ independent realizations of $X$ drawn from the target distribution $p_\pi(X)$, where $p_\mu(X) \neq p_\pi(X)$. Let the random variable $X \in \mathcal{X}$ represent the context (e.g. patient features) and let $T \in \mathcal{T}$ describe the intervention (treatment) assigned to the patient. Without loss of generality, consider the case when the treatment is binary, such that $\mathcal{T} = \{0, 1\}$. However, note that our model selection method is also applicable for any number of treatments. We use the potential outcomes framework (Rubin, 2005) to describe the result of performing an intervention $t \in \mathcal{T}$ as the potential outcome $Y(t) \in \mathcal{Y}$. Let $Y(1)$ represent the potential outcome under treatment and $Y(0)$ the potential outcome under control. Note that for each individual, we can only observe one of potential outcomes $Y(0)$ or $Y(1)$. We assume that the potential outcomes have a stationary distribution $p_\mu(Y(t) \mid X) = p_\pi(Y(t) \mid X)$ given the context $X$; this represents the *covariate shift* assumption in domain adaptation (Shimodaira, 2000).

Observational data can be used to estimate $\mathbb{E}[Y \mid X = x, T = t]$ through regression. Assumption 1 describes the causal identification conditions (Rosenbaum & Rubin, 1983), such that the potential outcomes are the same as the conditional expectation: $\mathbb{E}[Y(t) \mid X = x] = \mathbb{E}[Y \mid X = x, T = t]$.

**Assumption 1 (Consistency, Ignorability and Overlap).** *For any individual (unit) $i$, receiving treatment $t_i$, we observe $Y_i = Y(t_i)$. Moreover, $\{Y(0), Y(1)\}$ and the data generating process $p(X, T, Y)$ satisfy strong ignorability $Y(0), Y(1) \perp\!\!\!\perp T \mid X$ and overlap $\forall x : P(T \mid X = x) > 0$.*

The ignorability assumption, also known as the no hidden confounders (unconfoundedness) assumptions, means that we observe all variables $X$ that causally affect the assignment of the intervention and the outcome. Under unconfoundedness, $X$ blocks all backdoor paths between $Y$ and $A$ (Pearl, 2009).

Under Assumption 1, the conditional expectation of the potential outcomes can also be written as the interventional distribution obtained by applying the $do-$operator under the causal framework of Pearl (2009): $\mathbb{E}[Y(t) \mid X = x] = \mathbb{E}[Y \mid X = x, do(T = t)]$. This equivalence will enable us to reason

about causal graphs and interventions on causal graphs in the context of selecting ITE methods for estimating potential outcomes.

■ **Evaluating ITE models.** Methods for estimating ITE learn predictors $f : \mathcal{X} \times \mathcal{T} \to \mathcal{Y}$ such that $f(x, t)$ approximates $\mathbb{E}[Y \mid X = x, T = t] = \mathbb{E}[Y(t) \mid X = x] = \mathbb{E}[Y \mid X = x, do(T = t)]$. The goal is to estimate the ITE, also known as the conditional average treatment effect (CATE):

$$\tau(x) = \mathbb{E}[Y(1) \mid X = x] - \mathbb{E}[Y(0) \mid X = x] \tag{1}$$
$$= \mathbb{E}[Y \mid X = x, do(T = 1)] - \mathbb{E}[Y \mid X = x, do(T = 0)]. \tag{2}$$

The CATE is essential for individualized decision making as it guides treatment assignment policies. A trained ITE predictor $f(x, t)$ approximates CATE as: $\hat{\tau}(x) = f(x, 1) - f(x, 0)$. Commonly used to assess ITE models is the precision of estimating heterogeneous effects (PEHE) (Hill, 2011):

$$PEHE = \mathbb{E}_{x \sim p(x)}[(\tau(x) - \hat{\tau}(x))^2], \tag{3}$$

which quantifies a model's estimate of the heterogeneous treatment effects for patients in a population.

■ **UDA model selection.** Given a set $\mathcal{F} = \{f_1, \dots f_m\}$ of candidate ITE models trained on the source domain $\mathcal{D}_{src}$, our aim is to select the model that achieves the lowest target risk, that is the lowest PEHE on the target domain $\mathcal{D}_{tgt}$. Thus, ITE model selection for UDA involves finding:

$$\hat{f} = \underset{f \in F}{\arg\min} \, \mathbb{E}_{x \sim p_\pi(x)}[(\tau(x) - \hat{\tau}(x))^2] = \underset{f \in F}{\arg\min} \, \mathbb{E}_{x \sim p_\pi(x)}[(\tau(x) - (f(x, 1) - f(x, 0)))^2]. \tag{4}$$

For this purpose, we propose using the invariance of causal graphs across domains to select ITE predictors that are robust to distributional shifts in the marginal distribution of $X$.

## 3.2 CAUSAL GRAPHS FRAMEWORK

In this work, we use the semantic framework of causal graphs (Pearl, 2009) to reason about causality in the context of model selection. We assume that the unknown data generating process in the source domain can be described by the causal directed acyclic graph (DAG) $G$, which contains the relationships between the variables $V = (X, T, Y)$ consisting of the patient features $X$, treatment $T$, and outcome $Y$. We operate under the Markov and faithfulness conditions (Richardson, 2003; Pearl, 2009), meaning that any conditional independencies in the joint distribution of $p_\mu(X, T, Y)$ are indicated by $d$-separation in $G$ and vice-versa.

In this framework, an intervention on the treatment variable $T \in V$ is denoted through the do-operation $do(T = t)$ and induces the interventional DAG $G_{\overline{T}}$, where the edges into $T$ are removed. The interventional DAG $G_{\overline{T}}$ corresponds to the interventional distribution $p_\mu(X, Y \mid do(T = t))$ Pearl (2009). The only node on which we perform interventions in the target domain is the treatment node. Consequently, this node will have the edges into it removed, while the remainder of the DAG is unchanged. We assume that the causal graph is invariant across domains (Schoelkopf et al., 2012; Ghassami et al., 2017; Magliacane et al., 2018) which we formalize for interventions as follows:

**Assumption 2 (Causal invariance).** *Let $V = (X, T, Y)$ be a set of variables consisting of patient features $X$, treatment $T$, and outcome $Y$. Let $\Delta$ be a set of domains, $p_\delta(X, Y \mid do(T = t))$ be the corresponding interventional distribution on $V$ in domain $\delta \in \Delta$, and $I(p_\delta(V))$ denote the set of all conditional independence relationships embodied in $p_\delta(V)$, then*

$$\forall \delta_i, \delta_j \in \Delta, I(p_{\delta_i}(X, Y \mid do(T = t))) = I(p_{\delta_j}(X, Y \mid do(T = t))). \tag{5}$$

## 4 ITE MODEL SELECTION FOR UDA

Let $\mathcal{F} = \{f_1, f_2, \dots f_m\}$ be a set of candidate ITE models trained on the data from the source domain $\mathcal{D}_{src}$. Our aim is to select the model $f \in \mathcal{F}$ that achieves the lowest PEHE on the target domain $\mathcal{D}_{tgt}$, as described in Equation 4. Let $G$ be a causal graph, either known or discovered, that describes the causal relationships between the variables in $X$, the treatment $T$ and the outcome $Y$. Let $G_{\overline{T}}$ be the interventional causal graph of $G$ that has edges removed into the treatment variable $T$.

■ **Prior causal knowledge and graph discovery.** The invariant graph $G$ can be arrived at in two primary ways. The first would be through experimental means, such as randomized trials, which

does not scale to a large number of covariates due to financial or ethical impediments. The second would be through the causal discovery of DAG structure from observational data (for a listing of current algorithms we refer to (Glymour et al., 2019b)), which is more feasible in practice. Under the assumption of no hidden confounding variables, score-based causal discovery algorithms output a completed partially directed acyclical graph (CPDAG) representing the Markov equivalence class (MEC) of graphs, i.e., those graphs which are statistically indistinguishable given the observational data and therefore share the same conditional independencies. Provided a CPDAG, it is up to an expert (or further experiments) to orient any undirected edges of the CPDAG to convert it into the DAG (Pearl, 2009). This step is the most error-prone, and we show in our real data experiments how a subgraph (using only the known edges) can still improve model selection performance.

■ **Improving target risk estimation.** For the trained ITE model $f$, let $\hat{y}(0) = f(x, 0)$ and let $\hat{y}(1) = f(x, 1)$ be the predicted potential outcomes for $x \sim p_\pi(x)$. We develop a selection method that prefers models whose predictions on the target domain preserve the conditional independence relationships between $X, T$ and $Y$ in the interventional DAG $G_{\overline{T}}$ with edges removed into the treatment variable $T$. We first propose a Theorem, which we later exploit for model selection.

**Theorem 1.** *Let $p_\mu(X, T, Y)$ be a source distribution with corresponding DAG $G$. If $Y = f(X, T)$, i.e., $f$ is an optimal ITE model, then*

$$I_G(G_{\overline{T}}) = I(p_\pi(X, f(X, t) \mid do(T = t))), \tag{6}$$

*where $p_\pi(X, f(X, t) \mid do(T = t))$ is the interventional distribution for the target domain and $I_G(G_{\overline{T}})$ and $I(p_\pi(X, f(X, t) \mid do(T = t)))$ returns all the conditional independence relationships in $G_{\overline{T}}$ and $p_\pi(X, f(X, t) \mid do(T = t))$, respectively.*

For details and proof of Theorem 1 see Appendix B. Theorem 1 provides an equality relating the predictions of $f$ in the target domain to the interventional DAG $G_{\overline{T}}$. Therefore we desire the set of independence relationships in $G_{\overline{T}}$ to equal $I(p_\pi(X, f(X, t) \mid do(T = t)))$. In our case, we do not have access to the true interventional distribution $p_\pi(X, f(X, t) \mid do(T = t))$, but we can approximate it from the dataset obtained by augmenting the unlabeled target dataset $\mathcal{D}_{tgt}$ with the model's predictions of the potential outcomes: $\hat{\mathcal{D}}_{tgt} = \{(x_i^{tgt}, 0, \hat{y}_i^{tgt}(0)), (x_i^{tgt}, 1, \hat{y}_i^{tgt}(1))\}_{i=1}^{N_{tgt}}$, where $\hat{y}_i^{tgt}(t) = f(x_i^{tgt}, t)$, for $x_i^{tgt} \in \mathcal{D}_{tgt}$. We propose to improve the formalization in Eq. 4 by adding a constraint on preserving the conditional independencies of $G_{\overline{T}}$ as follows:

$$\underset{f \in F}{\arg\min} \, \mathcal{R}_T(f) \text{ s.t. } \mathbb{E}[NCI(G_{\overline{T}}, \hat{\mathcal{D}}_{tgt})] = 0, \tag{7}$$

where $\mathcal{R}_T(f)$ is a function that approximates the target risk for a model $f$, $NCI(G_{\overline{T}}, \hat{\mathcal{D}}_{tgt})$ is the number of conditional independence relationships in the graph $G_{\overline{T}}$ that are not satisfied by the test dataset augmented with the model's predictions of the potential outcomes $\hat{\mathcal{D}}_{tgt}$.

■ **Interventional causal model selection.** Consider the schematic in Figure 2. We propose an interventional causal model selection (ICMS) score that takes into account the model's risk on the source domain, but also the fitness to the interventional causal graph $G_{\overline{T}}$ on the target domain according to Eq. 4. A score that satisfies this is provided by the Lagrangian method:

$$\mathcal{L} = \mathcal{R}_T(f) + \lambda \mathbb{E}[NCI(G_{\overline{T}}, \hat{\mathcal{D}}_{tgt})]. \tag{8}$$

Figure 2: ICMS is unique in that it calculates a causal risk (green) using predictions on target data. Purple arrows denote pathways unique to ICMS.

The first term $\mathcal{R}_T(f)$ is equivalent to the expected test PEHE which at selection time can be approximated by the validation risk (either source or target risk), which we represent as $v_r(f, \mathcal{D}_v, \mathcal{D}_{tgt})$. The second term, $\mathbb{E}[NCI(G_{\overline{T}}, \hat{\mathcal{D}}_{tgt})]$, which is derived from Theorem 1, evaluates the number of conditional independence relationships resulting from $d$-separation in the graph $G_{\overline{T}}$ that are not satisfied by the test dataset augmented with the model's predictions of the potential outcomes $\hat{\mathcal{D}}_{tgt}$. However, this term may never equal 0 and directly minimizing $NCI(G_{\overline{T}}, \hat{\mathcal{D}}_{tgt})$ involves evaluating conditional independence relationships, which is a hard statistical problem, especially for continuous variables (Shah et al., 2020). Because of this we approximate it by using a causal fitness score

that measures the likelihood of a DAG given the augmented dataset $\mathcal{D}_{tgt}$, which we rewrite as $c_r(f, \mathcal{D}_{tgt}, G_{\overline{T}})$. This represents an alternative and equivalent approach, also used by score-based causal discovery methods (Ramsey et al., 2017b; Glymour et al., 2019c). Consider partitioning the source dataset $\mathcal{D}_{src} = \{(x_i^{src}, t_i^{src}, y_i^{src})\}_{i=1}^{N_{src}}$ into a training dataset $\mathcal{D}_{tr}$ and a validation dataset $\mathcal{D}_v$ such that $\mathcal{D}_{src} = \mathcal{D}_{tr} \cup \mathcal{D}_v$. From Eq. 8 we define our ICMS score $r$ as follows:

**Definition 1 (ICMS score).** *Let $f$ be an ITE predictor trained on $\mathcal{D}_{tr}$. Let $\mathcal{D}_{tgt} = \{(x_i^{tgt})\}_{i=1}^{N_{tgt}}$ be test dataset and let $G_{\overline{T}}$ be the interventional causal graph. We define the following selection score:*

$$r(f, \mathcal{D}_v, \mathcal{D}_{tgt}, G_{\overline{T}}) = v_r(f, \mathcal{D}_v, \mathcal{D}_{tgt}) + \lambda c_r(f, \mathcal{D}_{tgt}, G_{\overline{T}}) \tag{9}$$

*where $v_r$ measures the validation risk on the validation set $\mathcal{D}_v$ and $c_r$ is a scoring function, which we call causal risk, that measures the fitness of the interventional causal graph $G_{\overline{T}}$ to the dataset $\hat{\mathcal{D}}_{tgt} = \{(x_i^{tgt}, 0, \hat{y}_i^{tgt}(0)), (x_i^{tgt}, 1, \hat{y}_i^{tgt}(1))\}_{i=1}^{N_{tgt}}$, where $\hat{y}_i^{tgt}(t) = f(x_i^{tgt}, t)$, for $x_i^{tgt} \in \mathcal{D}_{tgt}$.*

The validation risk $v_r(f, \mathcal{D}_v, \mathcal{D}_{tgt})$ can either be (1) source risk where we use existing model selection scores for ITE (Alaa & van der Schaar, 2019; Van der Laan & Robins, 2003), or (2) an approximation of target risk using the preexisting methods of IWCV or DEV (Sugiyama et al., 2007; You et al., 2019). We describe in the following section how to compute the causal risk $c_r(f, \mathcal{D}_{tgt}, G_{\overline{T}})$. $\lambda$ is a tuning factor between our causal risk term and validation risk $v_r$. We currently set $\lambda = 1$ for our experiments, but ideally, $\lambda$ would be proportional to our certainty in our causal graph. We discuss alternative methods for selecting $\lambda$, as well as a $\lambda$ sensitivity analysis in Appendix F. We provide ICMS pseudocode and a graphical illustration for calculating ICMS in Appendix C. We provide additional practical considerations and experiments regarding computational complexity, a subgraph analysis, causal graph misspecifications, ICMS selection on tree-based methods, ICMS selection on causally invariant features, noisiness of fitness score, and additional further discussion in Appendix H.

■ **Assessing causal graph fitness.** The causal risk term $c_r(f, \mathcal{D}_{tgt}, G_{\overline{T}})$ as part of our ICMS score requires assessing the fitness of the dataset $\hat{\mathcal{D}}_{tgt}$ to the invariant causal knowledge in $G_{\overline{T}}$. Some options include noteworthy maximum-likelihood algorithms such as the Akaike Information Criterion (AIC) (Akaike, 1998) and Bayesian Information Criterion (BIC) (Schwarz, 1978). Both the BIC and AIC are penalized versions of the log-likelihood function of a DAG given data, e.g., $\mathcal{LL}(G_{\overline{T}} \mid \hat{\mathcal{D}}_{tgt})$. In score based causal discovery, the DAG that best fits the data will maximize the $\mathcal{LL}(G_{\overline{T}} \mid \hat{\mathcal{D}}_{tgt})$ subject to some model complexity penalty constraints. In this work, we are not searching between candidate causal graphs and only care about maximizing our DAG to dataset fitness. Thus, we use the negative log-likelihood of $G$ given $\hat{\mathcal{D}}_{tgt}$, i.e., $-\mathcal{LL}(G_{\overline{T}} \mid \hat{\mathcal{D}}_{tgt})$, for our *causal risk* term $c_r$. The $-\mathcal{LL}(G_{\overline{T}} \mid \hat{\mathcal{D}}_{tgt})$ has a smaller value when $G$ is closer to modeling the probability distribution in $\hat{\mathcal{D}}_{tgt}$, i.e., the predicted potential outcomes satisfy the conditional independence relationships in $G$.

In score-based causal discovery, the Bayesian Information Criterion (BIC) is a common score that is used to discover the completed partially directed acyclic graph (CPDAG), representing all DAGs in the MEC, from observational data. Under the Markov and faithfullness assumptions, every conditional independence in the MEC of $G$ is also in $\mathcal{D}$. The BIC score is defined as:

$$BIC(G|\mathcal{D}) = -LL(G|\mathcal{D}) + \left( \frac{\log_2 N}{2} \right) ||G||, \tag{10}$$

where $N$ is the data set size and $||G||$ is the dimensionality of $G$. For our function $f$ in Eq. 9, we use the BIC score. However, since $N$ and $||G||$ are held constant in our proposed method our function $f \propto -LL(G|\mathcal{D})$. To find the $LL(G|\mathcal{D})$ we use the following decomposition:

$$LL(G|\mathcal{D}) = -N \sum_{X_i PA_i} H_{\mathcal{D}}(X_i|PA_i), \tag{11}$$

where $N$ is the dataset size, $PA_i$ are the parent nodes of $X_i$ in $G$, and $H$ is the conditional entropy function which is given by (Darwiche, 2009) for discrete variables and by (Ross, 2014) for continuous or mixed variables.

■ **Limitations of UDA selection methods** In the ideal scenario, we would be able to leverage labeled samples in the target domain to estimate the target risk of a machine learning model. We can express the target risk $\mathcal{R}_{tgt}$ in terms of the testing loss as follows:

$$\mathcal{R}_{tgt} = \frac{1}{N_{tgt}} \sum ((Y^{tgt}(1) - Y^{tgt}(0)) - (f(x^{tgt}, 1) - f(x^{tgt}, 0))^2 \tag{12}$$

However, in general, we do not have access to the treatment responses for patients in the target set and, even if we did, we can only observe the factual outcome. Moreover, existing model selection methods for UDA only consider predictions on the source domain and do not take into account the predictions of the candidate model in the target domain. Specifically, DEV and IWCV calculate a density ratio or importance weight between the source and target domain as follows:

$$w_f(x) = \frac{p(d=1|x)}{p(d=0|x)} \frac{N^{src}}{N^{test}}, \tag{13}$$

where $d$ designates dataset domain (source is 0, target is 1), and $\frac{p(d=1|x)}{p(d=0|x)}$ can be estimated by a discriminative model to distinguish source from target samples (You et al., 2019). Both calculate their score as a function of $\Delta$ as follows:

$$\Delta = \frac{1}{N_v} \sum_{i=1}^{N_v} w_f(x_i^v) l(y_i^v, f(x_i^v, 0), f(x_i^v, 1)) \tag{14}$$

where $l(\cdot, \cdot, \cdot)$ is a validation loss, such as influence-function based validation (Alaa & van der Schaar, 2019). Note that the functions $l$ and $w$ are only defined in terms of validation features $x_i^v$ from the source dataset. Such selection scores can be used to compute the validation score $v_r(f, \mathcal{D}_v, \mathcal{D}_{tgt})$ part of the ICMS score.

However, our ICMS score also computes the likelihood of the interventional causal graph given the predictions of the model in the target domain as a proxy for the risk in the target domain. By adding the causal risk, we the improve the estimation of target risk. Additionally, we specifically make use of the estimated potential outcomes on the test set $f(x^{tgt}, 0)$ and $f(x^{tgt}, 1)$ to calculate our selection score as shown in Eq. 9. Fig. 2 depicts how we use the predictions of the target data to calculate our ICMS score.

## 5 EXPERIMENTS

We evaluate methods by the test performance in terms of the average PEHE of the top 10% of models in the list returned by the model selection benchmarks. We will refer to this as the PEHE-10 test error. We provide additional metrics for our results in Appendix G.1.

■ **Benchmark ITE models.** We show how the ICMS score improves model selection for state-of-the-art ITE methods based on neural networks: GANITE (Yoon et al., 2018), CFRNet (Johansson et al., 2018), TARNet (Johansson et al., 2018), SITE (Yao et al., 2018) and Gaussian processes: CMGP (Alaa & van der Schaar, 2017) and NSGP (Alaa & van der Schaar, 2018). These ITE methods use different techniques for estimating ITE and currently achieve the best performance on standard benchmark observational datasets (Alaa & van der Schaar, 2019). We iterate over each model multiple times and compare against various DAGs and held-out test sets. Having various DAG structures results in varying magnitudes of test error. Therefore, without changing the ranking of the models, we min-max normalize our test error between 0 and 1 for each DAG, such that equal weight is given to each experimental run, and a relative comparison across benchmark ITE models can be made.

■ **Benchmark methods.** We benchmark our proposed ITE model selection score ICMS against each of the following UDA selection methods developed for predictive models: IWCV (Long et al., 2018) and DEV (You et al., 2019). To approximate the source risk, i.e., the error of ITE methods in predicting potential outcomes on the source domain (validation set $\mathcal{D}_v$), we use the following standard ITE scores: MSE on the factual outcomes, inverse propensity weighted factual error (IPTW) (Van der Laan & Robins, 2003) and influence functions (IF) (Alaa & van der Schaar, 2019). Note that each score (MSE, IPTW, etc.) can be used to estimate the target risk in the UDA selection methods: IWCV, DEV, or ICMS. Specifically, we benchmark our method in conjunction with each combination of ITE model errors {MSE, IPTW, IF} with validation risk {∅, IWCV, DEV}. We include experiments with ∅, to demonstrate using source risk as an estimation of validation risk.

### 5.1 SYNTHETIC UDA MODEL SELECTION

■ **Data generation.** In this section, we evaluate our method in comparison to related selection methods on synthetic data. For each of the simulations, we generated a random DAG, $G$, with $n$

Table 1: PEHE-10 performance (with standard error) using ICMS on top of existing UDA methods. ICMS(■) means that the ■ was used as the validation risk $v_r$ in the ICMS. For example, ICMS(DEV(⋆)) represents DEV(⋆) selection used as the validation risk $v_r$ in the ICMS. The ⋆ indicates the method used to approximate the validation error on the source dataset. Our method (in bold) improves over each selection method over all models and source risk scores (Src.).

| SELECTION METHOD | GANITE | CFR | TAR | SITE | CMGP | NSGP |
|---|---|---|---|---|---|---|
| MSE | 0.395 (0.051) | 0.363 (0.042) | 0.391 (0.050) | 0.157 (0.035) | 0.131 (0.046) | 0.282 (0.049) |
| **ICMS(MSE)** | **0.222 (0.049)** | **0.212 (0.036)** | **0.264 (0.034)** | **0.126 (0.027)** | **0.120 (0.050)** | **0.210 (0.047)** |
| IWCV(MSE) | 0.348 (0.046) | 0.393 (0.044) | 0.364 (0.052) | 0.185 (0.033) | 0.201 (0.041) | 0.209 (0.040) |
| **ICMS(IWCV(MSE))** | **0.212 (0.043)** | **0.220 (0.051)** | **0.256 (0.039)** | **0.149 (0.033)** | **0.183 (0.055)** | **0.172 (0.043)** |
| DEV(MSE) | 0.398 (0.056) | 0.414 (0.042) | 0.427 (0.049) | 0.198 (0.038) | 0.239 (0.058) | 0.183 (0.048) |
| **ICMS(DEV(MSE))** | **0.224 (0.042)** | **0.210 (0.039)** | **0.269 (0.035)** | **0.120 (0.040)** | **0.160 (0.047)** | **0.160 (0.042)** |
| IPTW | 0.381 (0.049) | 0.355 (0.046) | 0.394 (0.052) | 0.357 (0.045) | 0.182 (0.046) | 0.292 (0.045) |
| **ICMS(IPTW)** | **0.220 (0.049)** | **0.217 (0.039)** | **0.272 (0.032)** | **0.228 (0.031)** | **0.140 (0.050)** | **0.207 (0.047)** |
| IWCV(IPTW) | 0.269 (0.055) | 0.518 (0.049) | 0.433 (0.038) | 0.416 (0.053) | 0.417 (0.043) | 0.475 (0.053) |
| **ICMS(IWCV(IPTW))** | **0.053 (0.028)** | **0.121 (0.034)** | **0.119 (0.035)** | **0.207 (0.039)** | **0.304 (0.059)** | **0.328 (0.058)** |
| DEV(IPTW) | 0.302 (0.072) | 0.472 (0.056) | 0.414 (0.049) | 0.400 (0.057) | 0.441 (0.071) | 0.493 (0.086) |
| **ICMS(DEV(IPTW))** | **0.087 (0.035)** | **0.194 (0.052)** | **0.120 (0.027)** | **0.220 (0.031)** | **0.282 (0.041)** | **0.355 (0.050)** |
| IF | 0.222 (0.041) | 0.255 (0.050) | 0.250 (0.046) | 0.321 (0.059) | 0.392 (0.051) | 0.376 (0.057) |
| **ICMS(IF)** | **0.127 (0.039)** | **0.166 (0.042)** | **0.190 (0.044)** | **0.215 (0.056)** | **0.212 (0.053)** | **0.250 (0.054)** |
| IWCV(IF) | 0.180 (0.059) | 0.364 (0.051) | 0.286 (0.041) | 0.293 (0.043) | 0.415 (0.048) | 0.437 (0.057) |
| **ICMS(IWCV(IF))** | **0.058 (0.018)** | **0.104 (0.025)** | **0.108 (0.033)** | **0.173 (0.028)** | **0.292 (0.062)** | **0.331 (0.051)** |
| DEV(IF) | 0.193 (0.058) | 0.415 (0.045) | 0.292 (0.046) | 0.214 (0.038) | 0.490 (0.043) | 0.544 (0.053) |
| **ICMS(DEV(IF))** | **0.069 (0.026)** | **0.191 (0.048)** | **0.107 (0.029)** | **0.147 (0.025)** | **0.229 (0.054)** | **0.364 (0.056)** |

vertices and up to $n(n-1)/2$ edges (the asymptotic maximum number of edges in a DAG) between them. We construct our datasets with functional relationships between variables with directed edges between them in $G$ and applied Gaussian noise (0 mean and 1 variance) to each. We provide further details and pseudocode in Appendix G.1. Using the structure of $G$, we synthesized 2000 samples for our observational source dataset $\mathcal{D}_{src}$. We randomly split $\mathcal{D}_{src}$ into a training set $\mathcal{D}_{tr}$ and validation set $\mathcal{D}_v$ with 80% and 20% of the samples, respectively. To generate the testing dataset $\mathcal{D}_{tgt}$, we use $G$ to generate 1000 samples where half of the dataset receives treatment, and the other half does not. For $\mathcal{D}_{tgt}$, we randomly shift the mean between 1 and 10 of at least one ancestor of $Y$ in $G$, whereas in $\mathcal{D}_{src}$ a mean of 0 is used. It is important to note that the actual outcome or response is never seen when selecting our models. Furthermore, the training dataset $\mathcal{D}_{src}$ is observational and contains selection bias into the treatment node, whereas the synthetic test set $\mathcal{D}_{tgt}$ does not, since it was generated by intervention at the treatment node. Our algorithm has only access to the covariates $X$ in $\mathcal{D}_{tgt}$.

■ **Improved selection for all ITE models.** Table 1 shows results of ICMS on synthetic data over the benchmark ITE models. Here, we evaluate three different types of selection baseline methods: MSE, IPTW, and IF. We then compare each baseline selection method with UDA methods: IWCV, DEV, and ICMS (proposed). We repeated the experiment over 50 different DAGs with 30 candidate models for each model architecture. Each of the candidate algorithms was trained using their published settings and hyperparameters, as detailed in Appendix E. In Table 1, we see that our proposed method (ICMS) improves on each baseline selection method by having a lower testing error in terms of PEHE-10 (and inversion count in Appendix G.1) over all treatment models.

## 5.2 APPLICATION TO THE COVID-19 RESPONSE

ICMS facilitates and improves model transfer across domains with disparate distributions, i.e., time, geographical location, etc., which we will demonstrate in this section for COVID-19. The COVID-19 pandemic challenged healthcare systems worldwide. At the peak of the outbreak, many countries experienced a shortage of life-saving equipment, such as ventilators and ICU beds. Considering data from the UK outbreak, the pandemic hit the urban population before spreading to the rural areas (Figure 3). This implies that if we reacted in a timely manner, we could transfer models trained on the urban population to the rural population. However, there is a significant domain shift as the rural population is older and has more preexisting conditions (Armstrong et al., 2020). Furthermore, at the time of model deployment in rural areas, there may be no labeled samples available. The characteristics of the two populations are summarized in Figure 3. We provide detailed dataset details and patient statistics in Appendix J.

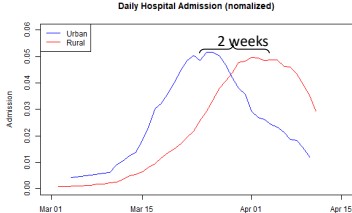 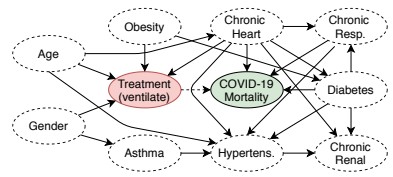

|  | Urban | Rural |
|---|---|---|
| Age (median) | 63 | 68 |
| Sex: male | 65% | 61% |
| Chronic Resp. Disease | 3.7% | 5.6% |
| Chronic Heart Disease | 3.5% | 8.1% |
| Obesity | 5.5% | 4.1% |
| Hypertension | 12.8% | 14.6% |

Figure 3: **Left:** COVID-19 pandemic hit urban areas before spreading to rural areas. **Middle:** Feature subset showing there exists a significant covariate shift between urban and rural populations with the urban population younger and with fewer preexisting conditions. **Right:** Discovered COVID-19 DAG.

■ **COVID-19 Ventilation UK (urban)** → **UK (rural).** Using the urban dataset, we performed causal discovery on the relationships between the patient covariates, treatment, and outcome. The discovered graph (Figure 3) agree well with the literature (Williamson et al., 2020; Niedzwiedz et al., 2020). To be able to evaluate the ITE methods on how well they estimate all counterfactual outcomes, we created a semi-synthetic version of the dataset with outcomes simulated according to the causal graph. Refer to Appendix J for details of the semi-synthetic data simulation. Our training observational dataset consists of the patient features, ventilator assignment (treatment) for the COVID-19 patients in the urban area, and the synthetic outcome generated based on the causal graph.

For each benchmark ITE model, we used 30 different hyperparameter settings and trained the various models to estimate the effect of ventilator use on the patient risk of mortality. We used the same training regime as in the synthetic experiments and the discovered COVID-19 causal DAG (using FGES Ramsey et al. (2017a)) shown in Figure 3. We evaluated the best ITE model selected by each model selection method in a ventilator assignment task. Using each selected ITE model, we assigned 2000 ventilators to the rural area patients that would have the highest estimated benefit (individualized treatment effect) from receiving the ventilator. Using the known synthetic outcomes for each patient, we then computed how many patients would have improved outcomes using each selected ITE model for assigning ventilators. By

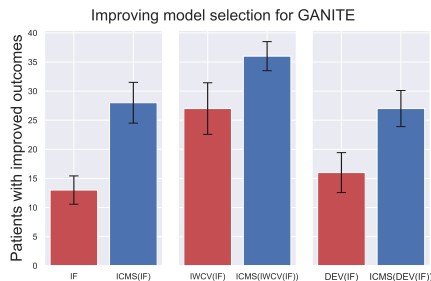

Figure 4: Performance of model selection methods in terms of the additional number of patients with improved outcomes compared to selecting models based on the factual error on the source domain.

considering selection based on the factual outcome (MSE) on the source dataset as a baseline, in Figure 4, we computed the additional number of patients with improved outcomes by using ICMS on top of existing UDA methods when selecting GANITE models with different settings of the hyperparameters. We see that ICMS (in blue) identified the GANITE models that resulted in better patient outcomes in the UK's rural areas without access to labeled data. We include additional experimental results in Appendix J.

■ **Additional experiments.** On the TWINS dataset (Almond et al., 2005) (in Appendix I), we show how our method improves UDA model selection even with partial knowledge of the causal graph (i.e., using only a known subgraph for computing the ICMS score). Note also that in the Twins dataset, we have access to real patient outcomes. Moreover, we also provide additional UDA model selection results for transferring domains on a prostate cancer dataset and the Infant Health and Development Program (IHDP) dataset (Hill, 2011) in Appendix I.

## 6 CONCLUSION

We provide a novel ITE model selection method for UDA that uniquely leverages the predictions of candidate models on a target domain by preserving invariant causal relationships. To the best of our knowledge, we have provided the first model selection method for ITE models specifically for UDA. We provide a theoretical justification for using ICMS and have shown on a variety of synthetic, semi-synthetic, and real data that our method can improve on existing state-of-the-art UDA methods.

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

## A  WHY USE CAUSAL GRAPHS FOR UDA?

To motivate our method, consider the following hypothetical scenario. Suppose we have $X_1$, $X_2$, $T$, and $Y$ representing age, respiratory comorbidities, treatment, and COVID-19 mortality, respectively, and the causal graph has structure $X_1 \rightarrow X_2 \rightarrow Y \leftarrow T$. Suppose that each node was a simple linear function of its predecessor with i.i.d. additive Gaussian noise terms. Now consider we have two countries $A$ and $B$, where $A$ has already been hit by COVID-19 and $B$ is just seeing cases increase (therefore have no observed outcomes yet). $B$ would like to select a machine learning model trained on the patient outcomes from $A$. However, $A$ and $B$ differ in distributions of age $X_1$. Consider the regression of $Y$ on $X_1$, $X_2$ and $T$, i.e., $Y = c_1 X_1 + c_2 X_2 + c_3 T$, by two models $f_1$ and $f_2$ that are fit on the source domain and evaluated on the target domain. Suppose that $f_1$ and $f_2$ have the same value for $c_2$ and $c_3$, but differ in $c_1$, where $c_1 = 0$ for $f_1$ and $c_1 \neq 0$ for $f_2$. We know that $Y$ is a function of only $X_1$ and $T$. Thus in the shifted test domain, $f_1$ must have a lower testing error than $f_2$, since the predictions of $f_2$ use $X_1$ (since $c_1 \neq 0$) and $f_1$ does not. Furthermore the predictions of $f_1$ have the same causal relationships and conditional independencies as $Y$, such as $f_1(X_1, X_2, T) \perp\!\!\!\perp X_2 \mid X_1$. This is not the case for $f_2$, where $f_2(X_1, X_2, T) \not\perp\!\!\!\perp X_2 \mid X_1$. Motivated by this, we can use a metric of graphical fitness of the predictions of $f_i$ to the underlying graphical structure to select models in shifted domains when all we have are unlabeled samples. As an added bonus, which we will highlight later, unlike existing UDA selection methods our method can be used without needing to share data between $A$ and $B$, which can help overcome patient privacy barriers that are ubiquitous in the healthcare setting.

## B  PROOF OF THEOREM 1

In this section, we present a proof for Theorem 1.

*Proof.* In the source domain, by the Markov and faithfullness assumptions the conditional independencies in $G$ are the same in $p_\mu(X, T, Y)$, such that

$$I_G(G) = I(p_\mu(X, T, Y)). \tag{15}$$

To estimate the potential outcomes $Y(t)$, we apply the $do$-operator to obtain the interventional DAG $G_{\overline{T}}$ and interventional distribution $p_\mu(X, Y \mid do(T = t))$, such that:

$$I_G(G_{\overline{T}}) = I(p_\mu(X, Y \mid do(T = t))). \tag{16}$$

Since we assume $Y = f(X, T)$ we obtain:

$$I_G(G_{\overline{T}}) = I(p_\mu(X, f(X, t) \mid do(T = t))). \tag{17}$$

By Assumption 2, we know that the conditional independence relationships in the interventional distribution are the same in any environment, so that

$$I(p_\mu(X, f(X, t) \mid do(T = t))) = I(p_\pi(X, f(X, t) \mid do(T = t))), \tag{18}$$

such that we obtain:

$$I_G(G_{\overline{T}}) = I(p_\pi(X, f(X, t) \mid do(T = t))). \tag{19}$$

$\square$

## C  ICMS ADDITIONAL DETAILS

To clarify our methodology further we have provided pseudocode in Algorithms 1 and 2. Algorithm 1 calculates the ICMS score (from Eq. 9) from a given model. The values for $c_r$ and $v_r$ are min-max normalized between 0 and 1 across all models. Algorithm 2 returns a ranked list of models by ICMS score from a set of ITE models $\mathcal{F}$. It takes optional prior knowledge in the form of a causal graph or known connections.

In Figure 5, we provide a graphical illustration for calculating $NCI$.

---

**Algorithm 1** Calculate ICMS

---

**Input:** ITE model $f$; source validation dataset $\mathcal{D}_v$; unlabeled target test set $\mathcal{D}_{tgt} = \{x_i^{tgt}\}_{i=1}^{N_{tgt}}$; interventional DAG $G_{\overline{T}}$; scale factor $\lambda$.
**Output:** ICMS score: $r(f, \mathcal{D}_v, \mathcal{D}_{tgt}, G_{\overline{T}})$
**Function:** ICMS($f, \mathcal{D}_v, \mathcal{D}_{tgt}, G_{\overline{T}}, \lambda$)**:**
$\hat{y}_i^{tgt}(t) \leftarrow f(x_i^{tgt}, t)$, for $x_i^{tgt} \in \mathcal{D}_{tgt}$
$\hat{\mathcal{D}}_{tgt} \leftarrow \{(x_i^{tgt}, 0, \hat{y}_i^{tgt}(0)), (x_i^{tgt}, 1, \hat{y}_i^{tgt}(1))\}_{i=1}^{N_{tgt}}$
$c_r \leftarrow$ Measure of $\hat{\mathcal{D}}_{tgt}$ to DAG $G_{\overline{T}}$ fitness.
$v_r \leftarrow$ Validation risk of $f$ on $\mathcal{D}_v$ and $\mathcal{D}_{tgt}$.
**return** $c_r + \lambda v_r$ (from Eq. 9).

---

**Algorithm 2** ICMS Selection

---

**Input:** Source dataset $\mathcal{D}_{src} = \{(x_i^{src}, t_i^{src}, y_i^{src})\}_{i=1}^{N_{src}}$ split into a training set $\mathcal{D}_{tr}$ and validation set $\mathcal{D}_v$; set of ITE models $\mathcal{F}$ trained $\mathcal{D}_{tr}$; unlabeled test set $\mathcal{D}_{tgt}$; optional prior knowledge in the form of a DAG $G_\pi$, scale factor $\lambda$.
**Output:** A list $\mathcal{F}'$ of models in $\mathcal{F}$ ranked by ICMS score.
**Function:** ICMS_sel($\mathcal{F}, \mathcal{D}_{tr}, \mathcal{D}_v, \mathcal{D}_{tgt}, \lambda, G_\pi = \emptyset$)**:**
$G_d \leftarrow$ causal discovery on $\mathcal{D}_{tr}$
$G \leftarrow$ assumed invariant DAG from $G_\pi$ or $G_d$
$G_{\overline{T}} \leftarrow$ interventional DAG of $G$ (remove edges into $T$)
$\mathcal{F}' \leftarrow$ Sort $\mathcal{F}$ by ICMS($f, \mathcal{D}_v, \mathcal{D}_{tgt}, G_{\overline{T}}, \lambda$) ascending
**return** $\mathcal{F}'$.

---

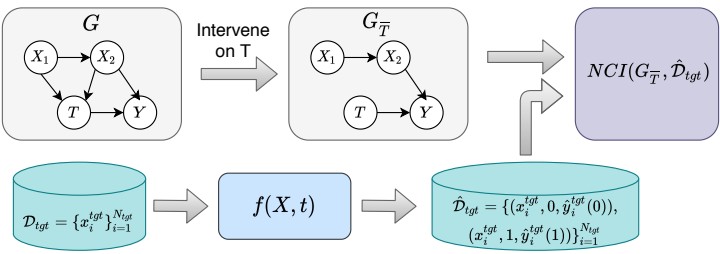

Figure 5: Schematic demonstrating calculation of $NCI$.

## D  CAUSAL DISCOVERY ALGORITHM DETAILS

In this section we discuss our causal discovery algorithms used. For real data, where we did not know all of the connections between variables, we discovered the remaining causal connections from the data using the Fast Greedy Equivalence Search (FGES) algorithm by (Ramsey et al., 2017a) on the entire dataset using the Tetrad software package (Glymour et al., 2019a). FGES assumes that all variables be observed and there is a linear Gaussian relationship between each node and its parent. Tetrad allows prior knowledge to be specified in terms of required edges that must exist, forbidden edges that will never exist, and temporal restrictions (variables that must precede other variables). Using our prior knowledge, we used the FGES algorithm in Tetrad to discover the causal DAGs for each of the public datasets. Only the directed edges that were output in the CPDAG by FGES were considered as known edges in the causal graphs. The Tetrad software package automatically handles continuous, discrete, and mixed connections, i.e., edges between discrete and continuous variables. If not using Tetrad for mixed variables, the method from (Ross, 2014) can be used.

# E    HYPERPARAMETERS FOR ITE MODELS

## E.1    GANITE

We used the publicly available implementation of GANITE[1], with the hyperparameters set as indicated in Table 2:

| Hyperparameter | Value |
|---|---|
| Optimization | Adam Moment Optimization |
| Batch size | 128 |
| $\alpha$ | 1 |
| Number of iterations | 5000 |
| Number of units per hidden layer | $s$ |
| Number of hidden layers | 2 |

Table 2: Hyperparameters used for GANITE. $s$ represents the number of input features.

## E.2    CFR AND TAR

For the implementation of CFR and TAR (Johansson et al., 2018), we used the publicly available code[2], with hyperameters set as described in Table 3. Note that for CFR we used Wasserstein regulatization, while for TAR the penalty imbalance parameter is set to 0.

| Hyperparameter | Value |
|---|---|
| Optimization | Adam Moment Optimization |
| Batch size | 100 |
| Num. of representation layers | 3 |
| Num. of hypothesis layers | 3 |
| Dim. of representation layers | 200 |
| Dim. of hypothesis layers | 100 |

Table 3: Hyperparameters used for CFR and TAR.

## E.3    SITE

For the implementation of SITE (Yao et al., 2018), we used the publicly available code[3], with hyperameters set as described in Table 4.

| Hyperparameter | Value |
|---|---|
| Optimization | Adam Moment Optimization |
| Batch size | 100 |
| Num. of representation layers | 3 |
| Num. of hypothesis layers | 3 |
| Dim. of representation layers | 200 |
| Dim. of hypothesis layers | 100 |
| $\lambda$ | $10^{-4}$ |

Table 4: Hyperparameters used for SITE.

---

[1] https://bitbucket.org/mvdschaar/mlforhealthlabpub/src/70a6f6130f90b7b2693505bb2f9ff78444541983/alg/ganite/
[2] https://github.com/clinicalml/cfrnet
[3] https://github.com/Osier-Yi/SITE

### E.4 CMGP AND NSGP

CMGP (Alaa et al., 2017) and NSGP (Alaa & van der Schaar, 2018) are ITE methods based on Gaussian Process models for which we used the publicly available implementation[4]. Note that for these ITE methods, the hyperparameters associated with the Gaussian Process are internally optimized.

## F LAMBDA

We base our choice of $\lambda$ to be proportional to our belief in our causal DAG that we use for UDA selection. If we are given prior knowledge in the form of a causal graph $G_\pi$. $G_\pi$ is optional and can be an empty graph as well. In either case we can use causal discovery on our observational dataset to discover a DAG $G_d$. Determining the edges that are truthful (and therefore invariant), in practice comes down to using human/expert knowledge to select the DAG that is most copacetic with existing beliefs of the natural world (Pearl, 2009). We refer to the selected truthful DAG as $G$, and we define $\lambda$ as follows:

$$\lambda = \frac{|E(G)|}{|E(G_\pi) \cup E(G_d)|}, \tag{20}$$

where $E(G)$ represents the set of edges of $G$ and $|E(G)|$ is the cardinality or number of edges in $G$. Intuitively, as the number of edges in our truthful dag $G$ decreases relative to our prior knowledge and what is discoverable from data, the less belief we have in our truth causal DAG. In the event that all causal edges are known ahead of time and is discoverable from data appropriately, then $\lambda = 1$.

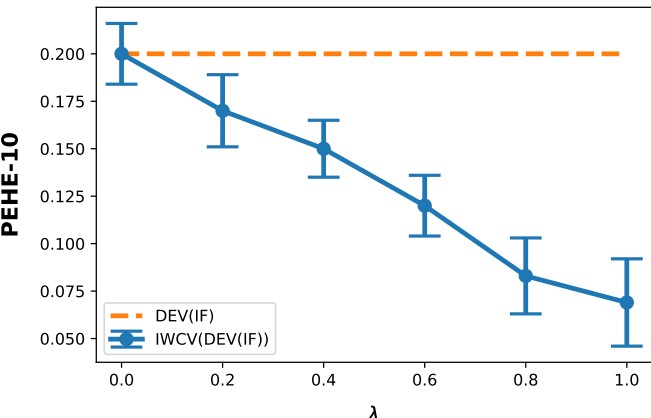

Figure 6: $\lambda$ sensitivity analysis.

■ **Lambda sensitivity.** We analyze the sensitivity of our method to the parameter $\lambda$ in Eq. 9. We used the same experimental set-up used for the synthetic experiments. Figure 6 shows the sensitivity of our method to $\lambda$ for GANITE using DEV and IF for calculating the validation risk $v_r$.

## G SYNTHETIC DATA GENERATION

Here we describe our synthetic data generation process (DGP). Algorithm 3 generates observational data according to a given invariant DAG $G$. Algorithm 4 generates interventional or treatment data according to a given invariant DAG $G$, where the treatment node is binarized and forced to have the value of 0 for half of the samples and 1 for the remainder.

---

[4] https://bitbucket.org/mvdschaar/mlforhealthlabpub/src/
70a6f6130f90b7b2693505bb2f9ff78444541983/alg/causal_multitask_gaussian_
processes_ite/

---

**Algorithm 3** Generate Observational Data

---

**Input:** A Graphical structure $G$, a mean $\mu$, standard deviation $\sigma$, edge weights $w$ and a dataset size $n$.
**Output:** An observation dataset according to $G$ with $n$ samples.
**Function:** `gen_obs_data`$(G, \mu, \sigma, w, n)$**:**
$e \leftarrow$ edges of $G$
$G_{sorted} \leftarrow$ `topological_sort`$(G)$
$ret \leftarrow$ empty list
**for** $node \in G$ **do**
   Append to $ret[node]$ a list of Gaussian ($\mu$ and $\sigma$) randomly sampled list of size $n$
**end for**
**for** $node \in G_{sorted}$ **do**
  **for** $par \in \{parents(node)\}$ **do**
    $ret[node] \mathrel{+}= ret[par] * w(par, node)$, where $w(par, node)$ is the edge weight from $par$ to $node$.
  **end for**
**end for**
Apply `sigmoid` function to the treatment node and binarize.
**return** $ret$.

---

### G.1 ADDITIONAL METRICS FOR SYNTHETIC EXPERIMENTS

We use an inversion count over the entire list of models, and provides a measure of list "sortedness". If we normalize this between the maximum number of inversions $n(n-1)/2$, where $n$ is the number of models in the list, then a completely sorted list in ascending order will have a value of 0. Similarly, a monotonically descending ordered list will have a value of 1. We provide additional synthetic results in terms of inversion count in Table 5.

Table 5: Inversion count using ICMS on top of existing UDA methods. ICMS(■) means that the ■ was used as the validation risk $v_r$ in the ICMS. For example, ICMS(DEV($\star$)) represents DEV($\star$) selection used as the validation risk $v_r$ in the ICMS. The $\star$ indicates the method used to approximate the validation error on the source dataset. Our method (in bold) improves over each selection method over all models and source risk scores (Src.).

| SELECTION METHOD | GANITE | CFR | TAR | SITE | CMGP | NSGP |
|---|---|---|---|---|---|---|
| MSE | 0.395 (0.071) | 0.363 (0.042) | 0.391 (0.050) | 0.157 (0.035) | 0.131 (0.046) | 0.282 (0.069) |
| **ICMS(MSE)** | **0.372 (0.069)** | **0.212 (0.036)** | **0.264 (0.034)** | **0.126 (0.027)** | **0.120 (0.050)** | **0.210 (0.067)** |
| IWCV(MSE) | 0.348 (0.056) | 0.393 (0.064) | 0.364 (0.052) | 0.185 (0.033) | 0.191 (0.081) | 0.209 (0.060) |
| **ICMS(IWCV(MSE))** | **0.352 (0.063)** | **0.220 (0.051)** | **0.256 (0.039)** | **0.149 (0.033)** | **0.183 (0.075)** | **0.172 (0.063)** |
| DEV(MSE) | 0.398 (0.076) | 0.414 (0.062) | 0.427 (0.049) | 0.198 (0.038) | 0.239 (0.078) | 0.163 (0.068) |
| **ICMS(DEV(MSE))** | **0.374 (0.062)** | **0.210 (0.049)** | **0.269 (0.035)** | **0.120 (0.040)** | **0.160 (0.067)** | **0.160 (0.062)** |
| IPTW | 0.395 (0.071) | 0.355 (0.046) | 0.391 (0.050) | 0.157 (0.035) | 0.182 (0.046) | 0.292 (0.075) |
| **ICMS(IPTW)** | **0.373 (0.069)** | **0.217 (0.039)** | **0.272 (0.032)** | **0.128 (0.031)** | **0.140 (0.050)** | **0.207 (0.067)** |
| IWCV(IPTW) | 0.269 (0.075) | 0.518 (0.059) | 0.433 (0.058) | 0.416 (0.053) | 0.417 (0.063) | 0.475 (0.083) |
| **ICMS(IWCV(IPTW))** | **0.073 (0.028)** | **0.121 (0.034)** | **0.119 (0.035)** | **0.207 (0.039)** | **0.304 (0.079)** | **0.328 (0.078)** |
| DEV(IPTW) | 0.302 (0.072) | 0.472 (0.056) | 0.414 (0.049) | 0.400 (0.057) | 0.441 (0.071) | 0.493 (0.086) |
| **ICMS(DEV(IPTW))** | **0.087 (0.035)** | **0.194 (0.052)** | **0.120 (0.027)** | **0.220 (0.031)** | **0.282 (0.041)** | **0.355 (0.077)** |
| IF | 0.222 (0.041) | 0.255 (0.050) | 0.250 (0.046) | 0.321 (0.059) | 0.392 (0.091) | 0.376 (0.097) |
| **ICMS(IF)** | **0.127 (0.039)** | **0.166 (0.042)** | **0.190 (0.044)** | **0.215 (0.076)** | **0.212 (0.073)** | **0.250 (0.084)** |
| IWCV(IF) | 0.18 (0.059) | 0.364 (0.051) | 0.286 (0.061) | 0.293 (0.043) | 0.415 (0.058) | 0.437 (0.087) |
| **ICMS(IWCV(IF))** | **0.058 (0.018)** | **0.104 (0.025)** | **0.108 (0.033)** | **0.173 (0.028)** | **0.292 (0.082)** | **0.331 (0.077)** |
| DEV(IF) | 0.193 (0.058) | 0.415 (0.075) | 0.292 (0.056) | 0.214 (0.038) | 0.490 (0.063) | 0.544 (0.093) |
| **ICMS(DEV(IF))** | **0.069 (0.026)** | **0.191 (0.048)** | **0.107 (0.029)** | **0.147 (0.025)** | **0.229 (0.074)** | **0.364 (0.076)** |

---

**Algorithm 4** Generate Treatment Data with perturbation

---

**Input:** A Graphical structure $G$, a mean $\mu$, standard deviation $\sigma$, edge weights $w$, a dataset size $n$, a list of perturbation nodes $p$, a perturbation mean $\mu_p$ and a perturbation standard deviation $\sigma_p$.
**Output:** An treatment dataset according to $G$ with $n$ samples and perturbation applied at nodes $p$.
**Function:** gen_treat_data($G, \mu, \sigma, w, n, \mu_p, \sigma_p$):
$e \leftarrow$ edges of $G$
$G_{sorted} \leftarrow$ topological_sort($G$)
$ret \leftarrow$ empty list
**for** $node \in G$ **do**
  **if** $node \in p$ **then**
    Append to $ret[node]$ a list of Gaussian ($\mu_p$ and $\sigma_p$) randomly sampled list of size $n$.
  **else**
    Append to $ret[node]$ a list of Gaussian ($\mu$ and $\sigma$) randomly sampled list of size $n$.
  **end if**
**end for**
**for** $node \in G_{sorted}$ **do**
  **for** $par \in \{parents(node)\}$ **do**
    **if** $node \notin$ treatment or response node **then**
      $ret[node] \mathrel{+}= ret[par] * w(par, node)$, where $w(par, node)$ is the edge weight from $par$
      to $node$.
    **end if**
  **end for**
**end for**
Binarize $ret[treat]$ into 50% with 0 value and the rest with 1 value.
$ret[response] \leftarrow$ incoming edges in $G$ multiplied by edge weights $w$.
**return** $ret$.

---

## H    PRACTICAL CONSIDERATIONS

Here we provide a discussion on some practical considerations.

■ **Computational complexity.** The computational complexity of ICMS as shown in Algorithm 1 and 2 scales linear with the number of models in $\mathcal{F}$. Specifically, the computational complexity is $\mathcal{O}(N_f \times Q(G, \mathcal{D}))$, where $N_f$ is the number of candidate models in $\mathcal{F}$ and $Q(G, \mathcal{D})$ is the computational complexity of calculating the fitness score of dataset $\mathcal{D}$ to $G$. In our case, we use the log-likelihood score, which requires calculating the conditional entropy between each parent node and child. In the worst case, this has a computational complexity of $\mathcal{O}(V_G^2)$, where $V_G$ is the number of vertices (or variables) in $G$ since a DAG with $V_G$ vertices will have an asymptotic number of edges $\frac{V_G(V_G-1)}{2}$.

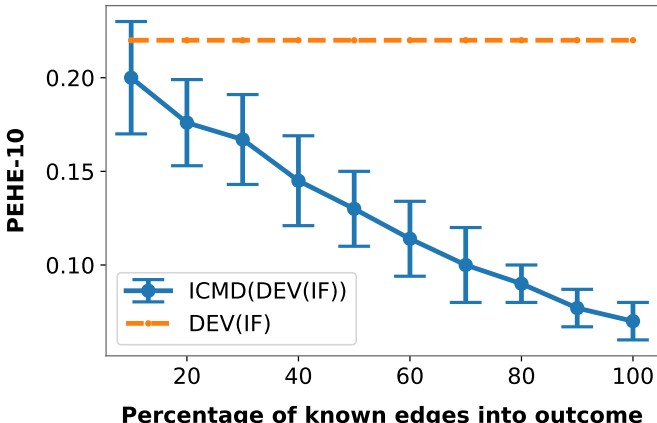

Figure 7: Performance gain in terms of known edges into the outcome node.

■ **Utilization of subgraphs.** In practice, we will likely not know the true underlying causal graph completely. Due to experimental, economical or ethical limitations, we often can not determine the orientation of all edges completely. Additionally, the process of causal discovery is not perfect and likely will result in unoriented, missing, or spurious edges that result from noisiness and biases in the observational dataset used. In Figure 7, we plot the performance of our ICMS method when selecting GANITE models as we increase the percentage of known edges into the outcome node in the causal subgraph used. We indeed prefer subgraphs that contain information about the parents of the outcome node. We conclude that it is perfectly admissible to use our methodology with a subgraph as input with the understanding that as edges are missing, performance degrades. However, the performance is still better than without using our ICMS score.

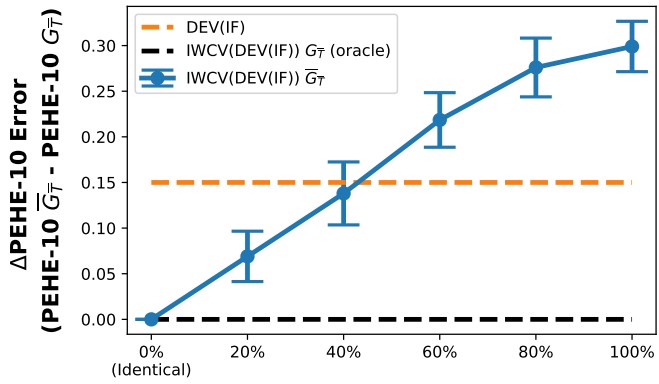

Figure 8: Performance of ICMS on incorrect graphs using IWCV(DEV(IF)). $\Delta$PEHE-10 error is the difference of the PEHE-10 error of $\overline{G_{\overline{T}}}$ and $G_{\overline{T}}$ using ICMS versus the percentage of graphical distance (in terms of total edges). $G_{\overline{T}}$ is the oracle causal graph and is held static across the $x$-axis.

■ **Analysis of causal graph correctness.** We investigate our method's sensitivity to incorrect causal knowledge. Here, we maliciously reverse or add spurious edges to our causal DAG when calculating ICMS. We used our same synthetic experimental setup, except we mutilate our oracle DAGs to form incorrect DAGs. We set $\lambda$ to 1 since we assume the graph is truth (even though it is incorrect). We use GANITE with DEV and IF as our validation risk metric and show our results in Fig. 8, which shows the $\Delta$PEHE-10 error, i.e., the difference in PEHE-10 error of the erroneous DAG $\overline{G_{\overline{T}}}$ and the oracle DAG $G_{\overline{T}}$, versus the percentage graph difference (between $G_{\overline{T}}$ and $\overline{G_{\overline{T}}}$). The graphical difference is calculated in terms of the percentage of edges that are mutated or removed. Fig. 8 shows the correlation between the correctness of the causal graph and the relative model selection improvement. This correlation testifies to the validity of ICMS, where a counterexample of our method would be incorrect DAGs leading to ICMS selecting better models (which is not the case).

■ **Noisiness of fitness score or graphs.** We would like to point out that there is noisiness in the fitness score that we use. The likelihood requires estimating the conditional entropy between each variable given their parents. This step is not perfect and there are many permutations of graphical structures that could have scores that are very close. We hypothesize that improving our fitness scores will likely improve the efficacy of our approach in general.

■ **Application: towards personalized model selection.** In some instances, various target domains may be represented by different underlying causal graphs (Shpitser & Sherman, 2018). Consider the following clinical scenario. Suppose that we have two target genetic populations A and B that each have their own unique causal graph. We have a large observational dataset with no genetic information about each patient. At inference time assuming that we know which genetic group a patient belongs to (and corresponding causal graph), we hypothesize that we can

Table 6: Additional PEHE-10 (with standard error) results for BART and Causal Forest using DEV and IF as validation risk.

| SEL. METHOD | BART | CSLFOREST |
|---|---|---|
| IF | 0.205 (0.032) | 0.253 (0.036) |
| **ICMS(IF)** | **0.098(0.030)** | **0.175(0.038)** |
| IWCV(IF) | 0.297 (0.039) | 0.288 (0.036) |
| **ICMS(IWCV(IF))** | **0.094(0.031)** | **0.189(0.029)** |
| DEV(IF) | 0.214 (0.036) | 0.308 (0.038) |
| **ICMS(DEV(IF))** | **0.082(0.023)** | **0.194(0.029)** |

select the models that will administer the more appropriate treatment for each genetic population using our proposed ICMS score.

■ **Tree-based methods.** Here we provide a brief experiment showing that ICMS improves over non-deep neural network approaches of Bayesian additive regression tree (BART) (Chipman et al., 2010) and Causal Forest (Wager & Athey, 2018) as well. Replicating our synthetic experiments, we evaluated BART and Causal Forest using ICMS with DEV, IWCV, and IF for a validation risk. In Table 6, we see that even for tree-based methods our ICMS metric is still able to select models that generalize best to the test domain.

■ **Model selection on causally invariant features.** Here we provide a brief experiment showing that ICMS can be used as a selection method for the causal feature selection algorithms of Rojas-Carulla et al. (2018); Magliacane et al. (2018). It is important to note that model selection is still important for models that are trained on an invariant set of causal features. These models can still converge to different local minima and have disparate performances on the target domain. Replicating our synthetic experiments, we used Rojas-Carulla et al. (2018) and Magliacane et al. (2018) to select causally in-

Table 7: Additional PEHE-10 (with standard error) results for Rojas-Carulla et al. (2018) (R.C. (2018)) and Magliacane et al. (2018) (Mag. (2018)) performance (with standard error) using DEV and IF as validation risk.

| Sel. Method | R.C. (2018) | Mag. (2018) |
|---|---|---|
| IF | 0.312 (0.033) | 0.381 (0.022) |
| **ICMS(IF)** | **0.192(0.021)** | **0.258(0.030)** |
| IWCV(IF) | 0.240 (0.029) | 0.292 (0.041) |
| **ICMS(IWCV(IF))** | **0.136(0.036)** | **0.183(0.037)** |
| DEV(IF) | 0.257 (0.025) | 0.212 (0.035) |
| **ICMS(DEV(IF))** | **0.110(0.024)** | **0.127(0.044)** |

variant features, which we use for training and testing our model. We then selected models using ICMS and compared against our standard benchmarks using GANITE. In Table 7, we see that even for these feature selection methods our ICMS metric is still able to select models that generalize best to the test domain (in comparison to DEV, IWCV, and IF).

# I EXPERIMENTAL SET-UP FOR SEMI-SYNTHETIC DATASETS AND ADDITIONAL RESULTS.

In this section, we highlight additional experiments performed on real datasets with semi-synthetic outcomes. Since real-world data rarely contains information about the ground truth causal effects, existing literature uses semi-synthetic datasets, where either the treatment or the outcome are simulated (Shalit et al., 2017). Thus, we evaluate our model selection method on a prostate cancer dataset and the IHDP dataset where the outcomes are simulated and on the Twins dataset (Almond et al., 2005) where the treatments are simulated. Furthermore, we provide UDA selection results on the prostate cancer dataset for factual outcomes as well.

Table 8: Results on IHDP, prostate cancer, and TWINS datasets. IF validation is used to compute the source risk. We report the PEHE-10 test error (with standard error) of various selections methods on ITE models. Our method (in bold) improves in terms of PEHE-10 over all methods and ITE models.

| DATASET | METHOD | GANITE | CFR | TAR | SITE | CMGP | NSGP |
|---|---|---|---|---|---|---|---|
| IHDP | IF | 0.186 (0.040) | 0.448 (0.052) | 0.444 (0.066) | 0.430 (0.050) | 0.461 (0.038) | 0.473 (0.066) |
| | **ICMS(IF)** | **0.105 (0.031)** | **0.386 (0.045)** | **0.246 (0.045)** | **0.342 (0.051)** | **0.380 (0.053)** | **0.462 (0.053)** |
| | IWCV(IF) | 0.134 (0.059) | 0.493 (0.055) | 0.412 (0.057) | 0.491 (0.057) | 0.519 (0.072) | 0.647 (0.090) |
| | **ICMS(IWCV(IF))** | **0.106 (0.023)** | **0.447 (0.036)** | **0.360 (0.047)** | **0.488 (0.073)** | **0.372 (0.095)** | **0.576 (0.019)** |
| | DEV | 0.174 (0.050) | 0.462 (0.036) | 0.403 (0.046) | 0.458 (0.043) | 0.550 (0.174) | 0.654 (0.097) |
| | **ICMS(DEV(IF))** | **0.095 (0.025)** | **0.438 (0.036)** | **0.427 (0.065)** | **0.405 (0.049)** | **0.475 (0.199)** | **0.583 (0.026)** |
| PC(UK)→SEER(US) | IF | 0.298 (0.053) | 0.377 (0.054) | 0.419 (0.054) | 0.194 (0.048) | 0.771 (0.042) | 0.679 (0.061) |
| | **ICMS(IF)** | **0.092 (0.057)** | **0.143 (0.026)** | **0.148 (0.054)** | **0.161 (0.039)** | **0.538 (0.028)** | **0.505 (0.051)** |
| | IWCV(IF) | 0.125 (0.058) | 0.340 (0.060) | 0.366 (0.035) | 0.398 (0.073) | 0.238 (0.051) | 0.481 (0.032) |
| | **ICMS(IWCV(IF))** | **0.018 (0.011)** | **0.146 (0.054)** | **0.218 (0.051)** | **0.331 (0.055)** | **0.161 (0.038)** | **0.329 (0.049)** |
| | DEV(IF) | 0.239 (0.068) | 0.308 (0.037) | 0.361 (0.064) | 0.348 (0.078) | 0.253 (0.065) | 0.480 (0.041) |
| | **ICMS(DEV(IF))** | **0.036 (0.013)** | **0.120 (0.038)** | **0.168 (0.057)** | **0.318 (0.062)** | **0.203 (0.032)** | **0.254 (0.057)** |
| TWINS→TWINS(SEMI) | IF | 0.286 (0.027) | 0.527 (0.054) | 0.464 (0.067) | 0.468 (0.102) | 0.223 (0.082) | 0.488 (0.087) |
| | **ICMS(IF)** | **0.193 (0.022)** | **0.370 (0.065)** | **0.309 (0.040)** | **0.299 (0.097)** | **0.152 (0.039)** | **0.164 (0.029)** |
| | IWCV(IF) | 0.495 (0.054) | 0.538 (0.051) | 0.574 (0.066) | 0.611 (0.075) | 0.438 (0.069) | 0.444 (0.077) |
| | **ICMS(IWCV(IF))** | **0.288 (0.059)** | **0.497 (0.074)** | **0.508 (0.048)** | **0.500 (0.077)** | **0.218 (0.055)** | **0.375 (0.099)** |
| | DEV(IF) | 0.435 (0.059) | 0.584 (0.084) | 0.518 (0.101) | 0.484 (0.115) | 0.351 (0.074) | 0.480 (0.089) |
| | **ICMS(DEV(IF))** | **0.277 (0.054)** | **0.512 (0.086)** | **0.475 (0.040)** | **0.447 (0.074)** | **0.227 (0.043)** | **0.411 (0.104)** |

Table 9: Results on predicting the outcome of prostate cancer given a treatment from models trained on the Prostate Cancer UK (PCUK) dataset and tested on the SEER dataset (United States). IF validation is used to compute the source risk. Here we show the factual error (of the top 10% of selected models) in terms of MSE of various selections methods on ITE models. Our method (in bold) improves in terms of test error over all methods and ITE models. The standard error is shown in parentheses.

| DATASET | METHOD | GANITE | CFR | TAR | SITE | CMGP | NSGP |
|---|---|---|---|---|---|---|---|
| PC(UK)→ PC(US) (REAL OUTCOMES) | IF | 0.256 (0.061) | 0.183 (0.078) | 0.319 (0.078) | 0.321 (0.013) | 0.305 (0.074) | 0.360 (0.082) |
| | **ICMS(IF)** | **0.108 (0.015)** | **0.127 (0.052)** | **0.311 (0.031)** | **0.243 (0.080)** | **0.258 (0.078)** | **0.294 (0.053)** |
| | IWCV(IF) | 0.280 (0.081) | 0.714 (0.061) | 0.595 (0.043) | 0.345 (0.051) | 0.297 (0.032) | 0.554 (0.057) |
| | **ICMS(IWCV(IF))** | **0.230 (0.014)** | **0.361 (0.035)** | **0.518 (0.049)** | **0.287 (0.037)** | **0.282 (0.042)** | **0.493 (0.019)** |
| | DEV(IF) | 0.231 (0.160) | 0.361 (0.129) | 0.448 (0.162) | 0.471 (0.172) | 0.379 (0.112) | 0.465 (0.163) |
| | **ICMS(DEV(IF))** | **0.123 (0.017)** | **0.313 (0.047)** | **0.396 (0.052)** | **0.326 (0.029)** | **0.332 (0.032)** | **0.412 (0.041)** |

■ **IHDP dataset.** The dataset was created by (Hill, 2011) from the Infant Health and Development Program (IHDP)[5] and contains information about the effects of specialist home visits on future cognitive scores. The dataset contains 747 samples (139 treated and 608 control) and 25 covariates about the children and their mothers. We use a set-up similar to the one in (Dorie et al., 2019) to simulate the outcome, while at the same time building the causal graph $G$.

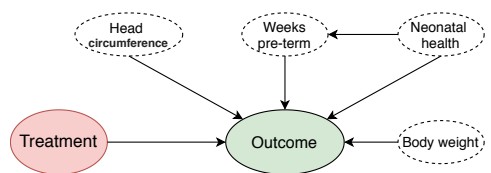

Figure 9: Interventional DAG for computing ICMS score on IHDP dataset.

[5]The dataset can be found as part of the Supplementary Files at https://www.tandfonline.com/doi/suppl/10.1198/jcgs.2010.08162?scroll=top

Since we do not have access to any real outcomes for this dataset, we build the DAG in Figure 9, such that a subset of the features affect the simulated outcome. Let $x$ represent the patient covariates and let $v$ be the covariates affecting the outcome in the DAG represented in Figure 9. We build the outcome for the treated patients $f(x, 1)$ and for the untreated patients $f(x, 0)$ as follows: $f(x, 0) = \exp(\beta(v + \frac{1}{2})) + \epsilon$ and $f(x, 1) = \beta v + \eta$ where $\beta$ consists of random regression coefficients uniformly sampled from $[0.1, 0.2, 0.3, 0.4]$ and $\epsilon \sim \mathcal{N}(0, 1)$, $\eta \sim \mathcal{N}(0, 1)$ are noise terms.

To create a target dataset with covariate shifts for the IHDP, we hold out the samples where the continuous variables neonatal health, head circumference and mom age have extreme values (either in the top 20% or the lowest 20%). We again ran 20 experiments and for each experiment we trained 30 candidate models for each model architecture. We use IF validation to approximate the source risk and we report the PEHE-10 test error. Table 8 illustrates the results on the IHDP dataset.

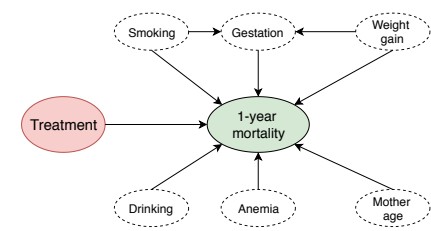

Figure 10: Interventional DAG for computing ICMS score on Twins dataset. The DAG contains a subset of the features available in the dataset for which we discovered causal relationships with the outcome indicated by the probability of 1-year mortality of the twin.

■ **TWINS dataset.** The TWINS dataset contains information about twin births in the US between 1989-1991 (Almond et al., 2005) [6]. The treatment $t = 1$ is defined as being the heavier twin and the outcome corresponds to the 1-year mortality. Since the dataset contains information about both twins we can consider their outcomes as being the potential outcomes for the treatment of being heavier at birth. The dataset consists of 11,400 pairs of twins and for each pair we have information about 30 variables related to their parents, pregnancy and birth.

We use the same set-up as in (Yoon et al., 2018) to create an observational study by selectively observing one of the twins based on their features (therefore inducing selection bias) as follows: $t \mid x \sim \text{Bernoulli}(\text{sigmoid}(w^T x + n))$ where $w \sim \mathcal{U}((-0.1, 0.1)^{30 \times 1})$ and $n \sim \mathcal{N}(0, 0.1)$.

Since we have access to the twins outcomes, we perform causal discovery to find causal relationships between the context features and the outcome. However, due to the fact that we do not have prior knowledge of the relationships between all 30 variables, we restrict the causal graph used to compute the causal risk to only contain a subset of variables, as illustrated in Figure 10.

Table 8 illustrates the results for the Twins dataset. Note that in this case, we use real outcomes and we also show the applicability of our method when only a subgraph of the true causal graph is known.

■ **Prostate cancer datasets.** In this case, we are a interested in deploying a machine learning model for prostate cancer but have access to only labeled data in the UK Biobank dataset, which has approximately 10,000 patients. We would like to deploy our models in the United States, where we have access to many samples

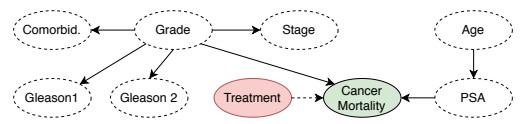

Figure 11: Interventional DAG for Prostate dataset.

of patient features, but no labeled outcome. For this target domain, we use the SEER dataset, which has over 100,000 samples. Our objective is to predict the patient mortality, given the patient features and treatment provided.

To be able to evaluate the methods on predicting counterfactual outcomes on the target domain (and thus compute the PEHE), we create a semi-synthetic dataset where the outcomes are simulated according to the discovered causal graph. Thus, we build the semi-synthetic outcomes for the prostate cancer dataset similarly to the IHDP dataset. Let $x$ represent the patient covariates and let $v$ be the covariates affecting the outcome. We build the outcome for the treated patients $f(x, 1)$ and for the untreated patients $f(x, 0)$ as follows: $f(x, 0) = \exp(\beta(v + \frac{1}{2})) + \epsilon$ and $f(x, 1) = \beta v + \eta$ where $\beta$ consists of random regression coefficients uniformly sampled from $[0.1, 0.2, 0.3, 0.4]$ and $\epsilon \sim \mathcal{N}(0, 0.1)$, $\eta \sim \mathcal{N}(0, 0.1)$ are noise terms.

---

[6] The data for the TWINS dataset can be found at `https://data.nber.org/data/linked-birth-infant-death-data-vital-statistics-data.html`

For the prostate cancer datasets, we also perform an experiment where we do not use semi-synthetic data (to generate the counterfactual outcomes), but use only the factual outcomes of the SEER dataset to evaluate our method. We train 30 models with identical hyperparameters as done in our synthetic and semi-synthetic experiments. We repeat this for all of our benchmark ITE methods. Table 9 shows that ICMS improves in terms of test error over all methods and ITE models.

■ **Computational settings.** All experiments were performed on an Ubuntu 18.04 system with 12 CPUs and 64 GB of RAM.

## J   COVID-19 EXPERIMENTAL DETAILS

### J.1   DATASET

We obtained de-identified COVID-19 Hospitalization in England Surveillance System (CHESS) data from Public Health England (PHE) for the period from 8[th] February (data collection start) to 14[th] April 2020, which contains 7,714 hospital admissions, including 3,092 ICU admissions from 94 NHS trusts across England. The data set features comprehensive information on patients' general health condition, COVID-19 specific risk factors (e.g., comorbidities), basic demographic information (age, sex, etc.), and tracks the entire patient treatment journey: hospitalization time, ICU admission, what treatment (e.g., ventilation) they received, and their outcome by April 20th, 2020 (609 deaths and 384 discharges). We split the data set into a source dataset containing 2,552 patients from urban areas (mostly Greater London area) and a target dataset of the remaining 5,162 rural patients.

### J.2   ABOUT THE CHESS DATA SET

COVID-19 Hospitalizations in England Surveillance System (CHESS) is a surveillance scheme for monitoring hospitalized COVID-19 patients. The scheme has been created in response to the rapidly evolving COVID-19 outbreak and has been developed by Public Health England (PHE). The scheme has been designed to monitor and estimate the impact of COVID-19 on the population in a timely fashion, to identify those who are most at risk and evaluate the effectiveness of countermeasures.

The CHESS data therefore captures information to fulfill the following objectives:

1. To monitor and estimate the impact of COVID-19 infection on the population, including estimating the proportion and rates of COVID-19 cases requiring hospitalisation and/or ICU/HDU admission
2. To describe the epidemiology of COVID-19 infection associated with hospital/ICU admission in terms of age, sex and underlying risk factors, and outcomes
3. To monitor pressures on acute health services
4. To inform transmission dynamic models to forecast healthcare burden and severity estimates

### J.3   COVID-19 PATIENT STATISTICS ACROSS GEOGRAPHICAL LOCATIONS

Figure 12 shows the histogram of age distribution for urban and rural patients. It is clear from the plot that the rural population is older, and therefore at higher risk of COVID-19. Table 10 presents statistics about the prevalence of preexisting medical conditions, the treatments received, and the final outcomes for patients in urban and rural areas. We can see that the rural patients tend to have more preexisting conditions such as chronic heart disease and hypertension. The higher prevalence's of comorbid conditions complicates the treatment for this population.

### J.4   DATA SIMULATION AND ADDITIONAL RESULTS USING ICMS

In the CHESS dataset, we only observe the factual patient outcomes. However, to be able to evaluate the selected ITE models on how well they estimate the treatment effects, we need to have access to both the factual and counterfactual outcomes. Thus, we have built a semi-synthetic version of

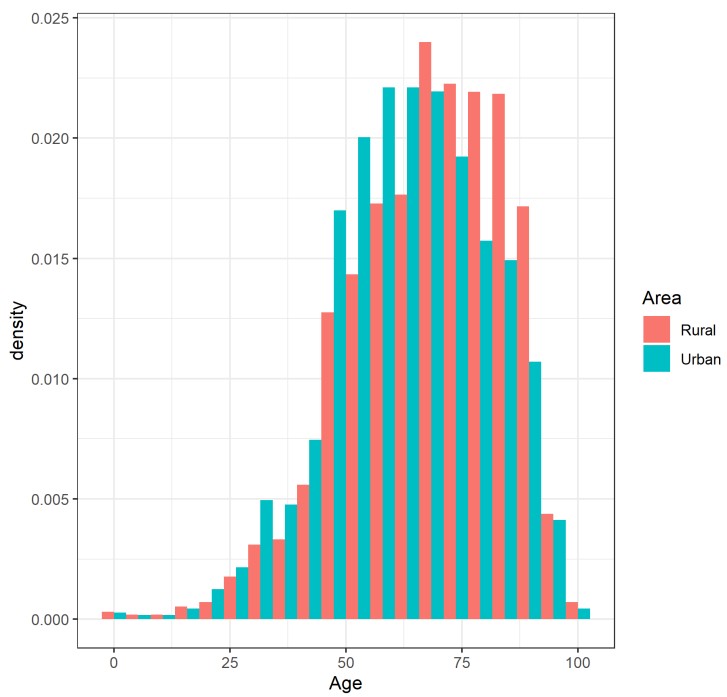

Figure 12: Age distribution for urban and rural patients. The median age of rural patients is five years older than the urban ones.

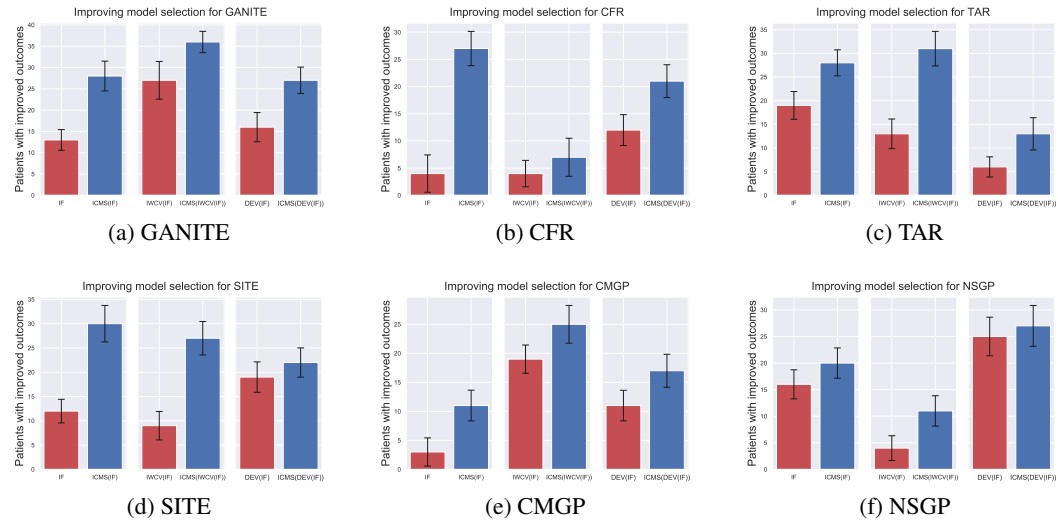

Figure 13: Performance of model selection methods in terms on additional number of patients with improved outcomes compared to selecting models based on the factual error on the source domain for all ITE models.

Table 10: Comparison of key features of urban and rural COVID-19 patients in the data set.

| | Urban | | Rural | |
| --- | --- | --- | --- | --- |
| | Percentage | Count | Percentage | Count |
| **Sex at Birth** | 65% | 1446 | 62% | 3388 |
| **Chonic Respiratory** | 4% | 81 | 6% | 310 |
| **Obesity** | 5% | 121 | 4% | 225 |
| **Chronic Heart** | 4% | 80 | 8% | 444 |
| **Hypertension** | 13% | 285 | 15% | 798 |
| **Asthma** | 4% | 92 | 6% | 326 |
| **Diabetes** | 9% | 197 | 11% | 589 |
| **Chronic Renal** | 2% | 45 | 3% | 175 |
| **Noninvasive Ventilation** | 7% | 160 | 6% | 342 |
| **Invasive Ventilation** | 21% | 456 | 16% | 879 |
| **Death** | 18% | 402 | 19% | 1014 |
| **Discharge** | 12% | 276 | 21% | 1164 |

the dataset, with potential outcomes simulated according to the causal graph discovered for the COVID-19 patients in Figure 3.

Let $x$ represent the patient covariates and let $v$ be the covariates affecting the outcome in the DAG represented in Figure 3. Let $f(x, 1)$ be the outcome for the patients that have received the ventilator (treatment) and let $f(x, 0)$ be the outcome for the patients that have not received the ventilator. The outcomes are simulated as follows: $f(x, 0) = \beta v + \eta$ and $f(x, 1) = \exp(\beta v) - 1 + \epsilon$, where $\beta$ consists of random regression coefficients uniformly sampled from $[0.1, 0.2, 0.3, 0.4]$ and $\epsilon \sim \mathcal{N}(0, 0.1)$, $\eta \sim \mathcal{N}(0, 0.1)$ are noise terms. We consider that the patient survives if $f(x, t) > 0$, where $t \in \{0, 1\}$ indicates the treatment received.

Our training observational dataset consists of the patient features $x$, ventilator assignment (treatment) $t$ for the COVID-19 patients in the urban area and the synthetic outcome generated using $f(x, t)$. For evaluation, we use the set-up described in Section 5.2 for assigning ventilators to patients in the rural area based on their estimated treatment effects. In Figure 13, we indicate the additional number of patients with improved outcomes by using ICMS on top of existing UDA methods when selecting ITE models with different settings of the hyperparameters.

