# OpenReview forum: "Selecting Treatment Effects Models for Domain Adaptation Using Causal Knowledge"
_ICLR.cc/2021/Conference — Reject_

### Official Review · AnonReviewer1 · 2020-10-26
**Very interesting, novel idea yet maybe needs to be linked more to other methods and evaluated accordingly**

**Rating:** 4
**Confidence:** 4

**Review:**

Summary: the paper attacks the problem of model selection for individual treatement effect (ITE) models when the domain of learning and prediction differ. Proposal is to use causal consistency as an additional "regularizer" in existing domain adaptation (DA) model selection methods. The "regularizer" would be scoring to which extent replacing factual outcomes by their counterfactual predictions would preserve conditional independence relations (induced by the causal graph) in the prediction domain. Experiments on a variety of datasets are conducted to show the added performance induced by using the "regularizer".

Good points:
- novel and very creative idea
- excellent writing that introduces concepts as they are needed + illustrative figures (Fig. 1 and 2)
- rigorous exposition of assumptions - quite important in this kind of problems
- interesting, illustrative use-case on covid

Questions:
- Is the dataset on covid openly accessible ?
- It seems that all base models are deep i.e. involving a non-convex optimization problem that makes the model prone to initial solution. Have you taken the "choice" of the random seed into account in the same manner as the hyper-parameters to select the model ?

Points currently limiting the relevance of the paper:
- [impact] model selection is here studied in isolation to other related techniques, notably causal feature selection (e.g. https://arxiv.org/abs/1911.07147) - it seems to me that it thus limits notably the generality of the conclusions as I don't see why practitioners would use one technique without the other. It is a big question mark for me and I'd like to read the point of view of authors on that point.
- [relevance of baselines] In related works authors mention two methods that they place as "state of the art" for ITE model selection: Causal Assurance and Influence Functions. Even if they are not designed specifically for the DA use case why not include them in the experiments ? It is often the case in practice that methods not designed for a specific case are unexpectedly robust to more specific setting. In particular I think it would be most telling to compare an augmented/"regularized" method (e.g. DEV+ICMS) vs generic SOTA methods (CA or IF). Disclaimer: I don't know enough of these methods to evaluate how easy it would be to perform such a comparison.
- [metrics] The standard PEHE metric seems to have been replaced by PEHE-10. I can't seem to find the definition of this metric in the paper ? why not use the standard ?
- [reproducibility] I can't seem to find the hyper-parameter grid that was used for the different base models and among which the competing methods would choose the best candidate - even in supplementary ? To allow other researchers to independently reproduce your results this is just mandatory.
- [reproducibility2] I can't find details on your strategy for seeding the different models. How many random seeds ? are they considered as hyper-params or part of an inner optimization loop ?
- [insights on results] The provided result in Table 1 indicates that adding ICMS to DEV or IWCV uniformly improves model selection. It seems in a sense "too good to be true". Can you provide intuition on why it is so ? Could it be possible to check that using ICMS alone is not already a very strong baseline - or is it in combination only that it works ?

Points that could improve the paper (and my score):
- provide a self-contained formula of NLL directly when introducing it in Sec. 4
- give full hyper-param grid for all models in supplementary
- define PEHE-10
- clarify setting for random seeds
- add an experiment with the Causal Assurance and Influence Functions baselines
- add an experiment comparing best model selected by proposal (e.g. DEV ICMS) vs best model found by causal feature selection (e.g. Rojas-Carulla 2018 or Magliacane 2018)

Overall, it seems to me the work deserves a more rigorous evaluation part as the underlying idea is very interesting and seems to be a very strong addition to the causal toolbox for ITE modeling.

---

> ### Author Response · Authors · 2020-11-17
> **Reply for Reviewer 1**
>
> We thank the reviewer for the constructive comments which have helped us improve the manuscript. We hope to address (in this response) all of the points you mentioned that would improve your score.
>
> **COVID-19 Dataset**
>
> Unfortunately, at the moment, the COVID-19 dataset is not publicly available.  However, we would like to note that other datasets used in the experiments (IHDP and Twins) are publicly available datasets.  We chose to perform experiments on the COVID-19 dataset due to its current relevance and importance to society.
>
> **Additional experiments with tree-based ITE methods**
>
> To highlight the effectiveness of our method with a wider range of ITE models, we have performed additional experiments to also showcase the ability of ICMS to select among tree-based ITE methods, namely BART and CausalForest. These results were added in Appendix H (Table 6).
>
> Nevertheless, please note that in the original manuscript, we did not evaluate ICMS only on deep learning methods but also on CMGP and NSGP, which are methods based on Gaussian Processes. With these additional experiments, we hope to have shown that ICMS is able to select the best models across a variety of ITE models, including deep learning methods, methods based on Gaussian Processes and tree-based methods.
>
>
> **Model selection and causal feature selection**
>
> We would like to emphasize that feature selection is a different problem to ITE model selection.  Methods for causal feature selection, such as [1,2,3], cannot address the problem of ITE model selection. In particular, suppose that we do feature selection first and then train different randomly initialized ITE models on the resulting datasets with the subset of selected features. The candidate ITE models can still converge to different local minima and have different performances in the target domain. Thus, model selection is still needed to be able to select the best one.
>
> To show this in practice, we have conducted an experiment in Appendix H, indicating that feature selection can be performed ahead of time, and ICMS can be used to select the best models for improved performance.
>
> **Relevant state-of-the-art benchmarks**
>
> Thank you for the suggestion.  Please note that Influence functions (IF) [4] is already a benchmark in our experiments (refer to the lower third of Table 1). Note also that IF [4] is a model selection metric for causal inference that is not designed for the UDA setting and can only be used to evaluate the source risk.
>
> Moreover, we would like to clarify that Causal Assurance is not a model selection metric for ITE models, nor for unsupervised domain adaptation. Causal Assurance is a model selection metric that was designed for domain robustness in the predictive setting and also only examines the source risk. On the other hand, the causal risk part of ICMS takes into account the unlabeled target dataset available in the UDA setting and the potential outcomes estimated on this target dataset. Nevertheless, to address your concern, we have performed additional experiments to also compare ICMS against Causal Assurance (CA).
>
>
> ----------------------------------------------------------------------------------------------------------------------------------------------------
>
>  Select. Methd.   |     ---GANITE----    |    ------CFR------            |         ------TAR------        |       ------SITE------        |      -----CMGP----       |      -----NSGP-----
>
> ----------------------------------------------------------------------------------------------------------------------------------------------------
>
> Causal assure.     | 0.138 (0.048) | 0.325 (0.060) | 0.240 (0.041) | 0.306 (0.043) | 0.321 (0.084) | 0.674 (0.084)
>
> ICMS(DEV(IF))     | 0.069 (0.026) | 0.191 (0.048) | 0.107 (0.029) | 0.147 (0.025) | 0.229 (0.054) | 0.365 (0.056)
>
>  ----------------------------------------------------------------------------------------------------------------------------------------------------
> This table shows a comparison of CA with the last row of Table 1 in the revised manuscript.

---

> > ### Author Response · Authors · 2020-11-17
> > **Additional reply for Reviewer 1**
> >
> > **Hyper-parameters**
> >
> > Please note that, as mentioned in Section 5.1, for each ITE model, we used the published settings of the hyperparameters and trained 30 candidate models from random initialization.  We have now included in Appendix E details about the hyperparameters and references to the code used for each ITE model. For reproducibility, we will also release the code for our model selection method upon acceptance.
> >
> > **Seeding**
> >
> > We would like to clarify that for each model, we hold the hyperparameters constant, as mentioned in the previous response.  The variation of each model can be attributed to 1) the weight initializations, which are random normal, and 2) the stochasticity of the order with which samples are seen during training.  We could have varied hyperparameters such as layers, neurons, etc., but the variability of generalization performance was already significant enough.
> >
> > **Insights on results**
> >
> > ICMS improves upon all the benchmarks.  Although this may appear “too good to be true,” it really is not - but rather expected.  This was highlighted in Appendix A.2 (in the original manuscript), which we have now moved into the main body of the revised paper.  We will summarize the two primary pitfalls of existing UDA methods - which allow us to improve upon them.  First, existing UDA methods are not designed for ITE models, where only the factual outcome is observed.  Second, and more importantly, existing methods do not factor in the predictions of candidate models on the target domain; rather, they weight the density ratios between samples from the target and source domain (looking only at input covariates) to weight model predictions (on only the source domain).  Without examining model predictions on the target domain, two models can have identical target risk estimates yet provide differing predictions in the target domain. We highlighted this in a motivational example in Appendix A, justifying why exploiting the causal graph can reconcile this issue.
> >
> > Moreover, the source of gain can also be understood from the definition of the ICMS score: $r(f, D_{v}, D_{tgt}, G_{\overline{T}}) = v_r(f, D_{v}, D_{tgt}) + \lambda c_r(f, D_{tgt}, G_{\overline{T}})$. Note that ICMS is a model selection method that is designed to be used in conjunction with a validation risk $v_r$, which is what existing methods (e.g DEV and IWCV) compute. The gain of our methods comes from also taking into account the causal risk $c_r$ as part of the model selection.
> >
> > Thus, at the heart of ICMS is the causal graph score - in this case, BIC.  By definition, the BIC (or any causal score for that matter) does not provide a measurement of predictive error, but rather of causal structure coherency, i.e., how likely are the parameters (causal structure) given the data.  Because of this, it is important to include a measurement of predictive error in the calculation - see the optimization in Eq. (7-9).
> >
> > To further clarify that ICMS is not designed to be used alone, we have improved the notation of our experimental results - which may have been a point of confusion.  Specifically, we have updated the notation in our Tables and Figures of the revised manuscript to say that ICMS is a function of validation risk. For example, see the updated Table 1 in the manuscript, where instead of MSE + ICMS, we have changed the notation to ICMS(MSE).
> >
> >
> > **Formula for log-likelihood**
> >
> > Please note that the formula for Negative Log-likelihood was already provided in Appendix H (in the original manuscript), but we have moved it to Section 4 in the revised manuscript, as suggested, under “Assessing causal graph fitness”.  Thank you!
> >
> >
> > **References**
> >
> > [1] Kui Yu, Xianjie Guo, Lin Liu, Jiuyong Li, Hao Wang, Zhaolong Ling, Xindong Wu. Causality-based Feature Selection: Methods and Evaluations. https://arxiv.org/abs/1911.07147
> >
> > [2] Sara Magliacane, Thijs van Ommen, Tom Claassen, Stephan Bongers, Philip Versteeg, and Joris M Mooij. Domain adaptation by using causal inference to predict invariant conditional distributions. Neurips (2018).
> >
> > [3] Mateo Rojas-Carulla, Bernhard Schölkopf, Richard Turner, and Jonas Peters. Invariant models for causal transfer learning. Journal of Machine Learning Research, 19(36):1–34, 2018
> >
> > [4] Ahmed Alaa and Mihaela van der Schaar. Validating causal inference models via influence functions. In Proceedings of the 36th International Conference on Machine Learning, volume 97 of Proceedings of Machine Learning Research, pp.191–201, 2019.

---

> ### Author Response · Authors · 2020-11-21
> **Follow-up**
>
> We thank you again for your initial comments on our manuscript.
>
> Please let us know if the revised manuscript and responses have addressed your concerns.  Furthermore, if you have additional comments, we are eager to address them!
>
> Thank you again!

---

### Official Review · AnonReviewer2 · 2020-10-27
**Theoretically sound and good experimental results**

**Rating:** 6
**Confidence:** 3

**Review:**

##########################

Summary:

This paper proposes a novel interventional causal model selection (ICMS) score to select individualized treatment effects (ITE) models under the unsupervised domain adaption (UDA) setting. The problem is fundamentally challenging as counterfactual outcomes cannot be observed. The authors make an assumption that the underlying causal structure across domains remains unchanged when adapting the ITE models from one domain to another. The authors propose Theorem 1 that the conditional independence relationships in the interventional DAG are equal to that in the interventional distribution for the target domain, followed by augmenting the target domain data with the model's prediction of the potential outcomes. Then the model that generates the best augmented target data in the sense that matches best with the interventional DAG is selected. To access this fitness, authors use the negative log-likelihood of the interventional DAG given the augmented data on the target domain. Finally, results with both synthetic data and real-world COVID-19 dataset show improvement for all ITE models.



##########################

Pros:

1. This paper tackles an important and challenging task: adapting ITE models unsupervisedly with unobserved counterfactual outcomes.
2. This paper is overall theoretically sound. The core assumption of causal structure invariance makes sense to me and the experimental results show significant improvement for all SOTA ITE models used.
3. This paper is well organized and easy to follow.

Based on these, I tend to vote for acceptance.


##########################

Cons:

1. In Eq (9) and Fig. 2, is the validation risk $v_r$ computed using the source dataset? This seems unjustified given that the risk computed in the source domain might not be a good approximation of that computed in the target domain.
2. In Eq. (9) and the paragraph ``Assessing causal graph ﬁtness'', $c_r$ is computed by the negative log-likelihood of G given the augmented dataset Dhat_{tgt}. This fitness makes sense to me: the model that generates the augmented dataset in target domain that best align with $G_{\bar{T}}$ should be selected. However, what is the relationship with Eq. (7-8)? It does not seem straightforward to see how Eq (9) and the NCI term are linked up.

---

> ### Author Response · Authors · 2020-11-17
> **Reply for Reviewer 2**
>
> We thank the reviewer for the constructive comments which have helped us improve the manuscript.
>
> **Validation risk**
>
> We are operating in the UDA setting where we have labeled data in the source domain and **unlabeled data** in the target domain, which means that it is not possible to compute the risk on the target domain directly. The validation risk in Eq (9) and Fig. 2 does use both the source (labeled) and target (unlabeled) datasets. In particular, to approximate the target risk, IWCV and DEV calculate density ratios between the covariates in the source and target dataset, which are used as importance weights in their metric. We believe this confusion comes from the fact that in the original manuscript, we did not include the formulas used for DEV/IWCV (which were originally in the Appendix). We have now moved that section into the body of the revised manuscript. Please refer to Section 4. We hope that this provides additional clarity.
>
>
> **Assessing causal graph fitness**
>
> Thank you for pointing this out. Let us clarify the relationship between NCI in Eq. (7-8) and $c_r$ in Eq. (9). $NCI(G_{\overline{T}}, D_{tgt})$ represents the number of conditional independence relationships resulting from d-separation in the graph $G_{\overline{T}}$ that are not satisfied by the test dataset augmented with the model's predictions of the potential outcomes $D_{tgt}$. This objective provides a theoretical justification for our model selection method and is derived from Theorem 1. However, directly minimizing NCI involves evaluating conditional independence relationships, which is a hard statistical problem, especially for continuous variables [1]. Because of this, we use the negative log-likelihood to evaluate the dataset fitness to the causal graph [2], which is an alternative and equivalent approach, also used by score based causal discovery methods [3,4]. We have further clarified this equivalence in the revised paper in Section 4.Note that, in our method, the interventional DAG $G_{\overline{T}}$ is held constant, and $D_{tgt}$ changes. Since the DAG is held constant, the penalty term for model complexity in the BIC score can be ignored, which leaves us with the negative log-likelihood of the DAG given data.
>
>
> **References**
>
> [1] Shah, Rajen D., and Jonas Peters. "The hardness of conditional independence testing and the generalised covariance measure." Annals of Statistics 48.3 (2020): 1514-1538.
>
> [2] Pearl, Judea. “Causality: Models, Reasoning, and Inference.” 2009.
>
> [3] Joseph Ramsey, Madelyn Glymour, Ruben Sanchez-Romero, and Clark Glymour. “A million variables and more: the fast greedy equivalence search algorithm for learning high-dimensional graphical causal models, with an application to functional magnetic resonance images.” International Journal of  Data  Science and  Analytics,  3(2):121–129,  Mar  2017.
>
> [4] Peter Spirtes, Kun Zhang, Peter Spirtes. “Review of Causal Discovery Methods Based on Graphical Models.”  Frontiers in Genetics (2019).

---

### Official Review · AnonReviewer4 · 2020-10-27
**A very well-written paper with incremental innovation**

**Rating:** 6
**Confidence:** 5

**Review:**

The paper introduces a model selection metric for ITE models under the unsupervised domain adaptation setting. By modeling the data from a source domain, the metric is able to select optimal ITE models that perform well on a target domain, where only unlabeled data available. Experiments are conducted on both synthetic and real-world datasets.

+ The problem is significant in machine learning and the idea is incrementally innovative: causal inference under UDA setting.
+ The idea flow is logical. The paper is very well-written.

Questions:
1. How to evaluate the generated causal graph? The causal structure learned from the source domain is used to estimate the causal risk on the target domain. How to evaluate the goodness of a learned causal structure? Especially in the real-world dataset, the causal graph could be much more complex than the synthetic setting. How to evaluate the dependencies among each pair of nodes?

2. How to evaluate the parameter lambda? In equation (8), a parameter lambda is used for controlling the weight of NCI and set to 1. Additional experiments should be conducted to evaluate the performance under different values of lambda.

3. Some notations are not clear.
- In equation (7), the definition of NCI is not clear. A graphic illustration of NCI results in the causal graph should be provided.

---

> ### Author Response · Authors · 2020-11-17
> **Reply for Reviewer 4**
>
> We thank the reviewer for the constructive comments which have helped us improve the manuscript.
>
> **Causal graph evaluation**
>
> In this work, the causal graph can be obtained via a causal discovery algorithm. By definition, the discovered graph belongs to the Markov Equivalence Class (MEC) of graphs that are statistically indistinguishable from each other, given the observational data [1,3]. Because the graph was found using a score-based causal discovery algorithm (the FGES algorithm), the graph is optimal within the MEC - and by definition, has maximized the graphical fitness score, such as the Bayesian Information Criterion (BIC). A human expert can, however, help to augment the discovery step by obligating certain edge constraints while performing the causal discovery step [1,3].
>
> The goodness of the fit can only be determined on the source domain since we only have labels for this domain. This would be done by calculating the BIC score of the discovered graph given the observational data from the source domain. Note that by the Markov and Faithfulness assumptions that this implies that every edge in $G$ represents a causal dependency in the probability distribution, and vice-versa.
>
> We would also like to emphasize that causal discovery is not the main focus of our work, and we assume that the causal graph is given. While we describe in the paper in Section 4 the practical considerations for obtaining a causal graph, the novelty of our method comes from incorporating such causal knowledge into a model selection method for ITE models in the UDA setting. Nevertheless, we acknowledge that our framework relies on having at least partially correct causal knowledge and in our experiments, we have also investigated the robustness of ICMS to incomplete causal knowledge (see Figure 7) and to misspecified edges in the causal graph (see Figure 8) used for computing the causal risk score.
>
> **Lambda parameter**
>
> We thank the reviewer for the comment, and we agree that the setting of the lambda parameter is important. To further highlight this aspect, we have performed additional experiments in Appendix F in the revised manuscript, where we show the sensitivity of our method to the setting of the lambda parameter.
>
> **Graphical illustration of NCI**
> $NCI(G_{\overline{T}}, D_{tgt})$ represents the number of conditional independence relationships resulting from d-separation in the graph $G_{\overline{T}}$ that are not satisfied by the test dataset augmented with the model's predictions of the potential outcomes $D_{tgt}$. To clarify how NCI could be calculated, we have provided a schematic in Figure 5 and demonstrates how $D_{tgt}$ is generated. Since in some cases the NCI term may never equal 0 and testing for conditional independence is a hard statistical problem, especially for continuous variables [2], we propose a different approach for evaluating the fitness of $D_{tgt}$ to the causal graph $G_{\overline{T}}$. In particular, we evaluate the causal risk using a causal fitness score that measures the likelihood of a DAG given some data on the test dataset, which we rewrite as $c_r(f, D_{tgt}, G_{\overline{T}})$. Thank you for the suggestion. We have included additional explanations about the NCI in Section 4 of the paper.
>
> **References**
>
> [1] Pearl, Judea. “Causality: Models, Reasoning, and Inference.” 2009.
>
> [2] Shah, Rajen D., and Jonas Peters. "The hardness of conditional independence testing and the generalised covariance measure." Annals of Statistics 48.3 (2020): 1514-1538.
>
> [3] Peter Spirtes, Kun Zhang, Peter Spirtes. “Review of Causal Discovery Methods Based on Graphical Models.”  Frontiers in Genetics (2019).

---

### Official Review · AnonReviewer3 · 2020-10-28
**Interesting novel approach combining treatment effect estimation and domain adaptation, leveraging the concept of causal invariances.**

**Rating:** 8
**Confidence:** 4

**Review:**

Summary:
The present paper proposes a novel approach for model selection for individual treatment effect (ITE) estimation in the unsupervised domain adaptation (UDA) setting. The motivation for this approach, called interventional causal model selection (ICMS), is to exploit causal invariance to choose a model that has both good predictive power and fits the a priori belief for causal relationships.
More precisely, the authors prove a necessary condition for optimality of an ITE model and combine this condition with the classical target risk minimization from UDA model selection.
This necessary condition states preservation of all conditional independencies of the causal DAG by the interventional distribution of the target domain.

Recommendation:
Clear accept. The framework proposed in this work constitutes an interesting new approach to model selection exploiting ideas from causal invariance and also allows to adapt existing ITE estimation methods to tackle covariate shift problems.

Strong points:
 - This framework is theoretically motivated, using the structural causal model framework, and allows to improve various methods by allowing for model selection in UDA setting, as illustrated on a wide range of examples (simulations and real data).
 - The flexibility also extends to the quality of the data or prior knowledge, as this framework allows to incorporate as much expert structure knowledge as possible, or to specify its importance in the model selection via the hyperparameter $\lambda$.

Weak points:
 - The role of the causal DAG is key to this approach, it is however difficult to assess from the experiments and discussion, how much the proposed framework relies on the correctness of the used DAG, or, put differently, how sensitive this framework is to mis-specifications of the DAG.
 - It would be interesting to see the code for the presented examples and also to test it on other applications. Will it be made public at some point?
 - The presented work is considerable and it many experiments are provided. However I think that the experiments section is too long compared to the main section (Section 4). For instance, (parts of) the appendix A could be moved into the main part while shortening the Section 5.2.

Questions/Issues:
 - Sec. 5.2.: Reference for FGES is missing in the main part of the article (it's in the appendix but should also be in cited in Section 5.2)
 - Appendix B (additional related work) should be moved to the main part (Section 2).
 - Sec. 5.1: What is the reason for only shifting one ancestor of $Y$ in $G$? And can it also be, at the same time, an ancestor of the treatment, i.e., a confounder in the observational data (I think yes, since the intervention on the treatment variable cuts of incoming arrows, but has this been tested in practice as well?)?
 - Sec. 5.1: How do the authors choose the 30 candidate models for each model architecture? Is the variability across the different models similar for all comparable for all compared architecture?

Minor comments (that did not impact the score):
 - p. 3: unconfoundness $>>$ unconfoundedness
 - p. 6: i.e $>>$ missing comma (i.e.,)
 - p. 6: return $>>$ returned
 - p. 7: For completeness, add reference for domain shift between rural and urban populations.
 - p. 7: hospital admission $>>$ hospital admissions

---

> ### Author Response · Authors · 2020-11-17
> **Reply for Reviewer 3**
>
> We thank the reviewer for the constructive comments which have helped us improve the manuscript.
>
> **Sensitivity of our framework to incompleteness/misspecification of causal DAG**
>
> Thank you very much for your important point regarding the robustness of ICMS to incomplete/misspecified causal DAGs. We would first like to highlight that in Figure 5 of the original manuscript (Figure 7 in the revised manuscript), we provided an analysis of the performance gains of using ICMS as a selection metric in terms of the percentage of the known edges into the outcome in the causal graph used to compute the causal risk. Additionally, the experiments on the semi-synthetic Twins dataset from Appendix I also use an incomplete causal graph to compute the causal risk.
>
> To further highlight our method's robustness to the misspecifications of the causal DAG, we have performed additional experiments. Please refer to the new Figure 8 (Appendix H) in the revised manuscript, which shows how the ability of ICMS to select the best performing models decreases when increasing the number of misspecified edges in the causal graph used to compute the causal risk. Note that this is, in fact, a testament to the efficacy of our method, i.e., if our method did not show this behavior, then that would imply the causal structure did not matter. We thank the reviewer for the comment.
>
> **Code availability**
>
> Our code will be made publicly available upon acceptance.
>
> **Lengthy experiments section and manuscript layout**
>
> In the revised manuscript, as the reviewer suggested, we have condensed our experimental section by offloading the COVID-19 dataset details into Appendix J. Furthermore, we have moved Appendix A.2 (Limitations of Existing UDA methods) into Section 4 (methodology), as well as Appendix B into Section 2 (Related Works). We thank the reviewer for the comment that helps us to improve the paper clarity.
>
> **FGES reference**
>
> We have added in the FGES reference to Section 5.2. Thank you for pointing this out.
>
> **Sec 5.1 shifting one ancestor of Y in G**
>
> To create a covariate shift between the source and target domains, we shifted **at least** one ancestor node of Y in G. Furthermore, you are indeed correct. The nodes that are shifted can also be parent notes of the treatment, i.e., confounders in the observational data, and our experimental results already include this scenario. However, note that our model selection method is using the interventional DAG $G_{\overline{T}}$, with incoming edges into $T$ removed to compute the causal risk on the target domain and is agnostic to such selection bias in the target domain. We have clarified this in the revised manuscript.
>
> **Sec 5.1 candidate models**
>
> We thank the reviewer for the comment. For each ITE model, we used the published settings of the hyperparameters and trained 30 candidate models from random initialization. In particular, the variability is controlled (and standardized) across all candidate models since we only allowed random weight instantiations and stochastic gradient descent to determine the variation in the trained models. We have added a new Appendix E to describe the hyperparameter settings used for each ITE model and reference the publicly available implementations of the ITE methods from which the hyperparameter settings were obtained.
>
> **Typos and minor comments**
>
> We thank the reviewer for pointing out the syntax errors. We have incorporated the suggested corrections in the revised manuscript.

---

### Author Response · Authors · 2020-11-17
**Revised paper**

We thank all reviewers for the time and effort spent reviewing and commenting on our manuscript.  All of your inputs have been considered, and have helped to improve the exposition and rigor of ICMS.

We have updated the paper to incorporate the constructive feedback provided by the reviewers.  We have indicated our revisions in the revised manuscript by green text.  Please see individual responses to reviewers for revision details.

Here we provide a brief summary of revisions to the manuscript:
1) We have added in many new experiments: a) sensitivity of ICMS to misspecifications of causal DAG, b) sensitivity of parameter Lambda, c) evaluation of ICMS using tree-based ITE methods, and d) selection of models using ICMS on top of causal feature-selection methods.
2) We have moved Appendix A.2 (Limitations of Existing UDA methods) into Section 4 (methodology), as well as Appendix B into Section 2 (Related Works).
3) We have added in a graphical illustration of NCI as well as discussed the connection of NCI to log-likelihood.
4) We have specified all hyperparameters used for each ITE method in Appendix E.
5) We have clarified our notations in our Tables and Figures to indicate that ICMS is a function of validation risk.  For example, DEV(IF) + ICMS has been changed to ICMS(DEV(IF)).

Please let us know if further clarifications are needed.

---

### Decision · Program_Chairs · 2021-01-07
**Final Decision**

**Decision:**

Reject

**Comment:**

This paper considers the problem of identification of causal effects under the unsupervised domain adaptation setting. The authors assume the invariance of the causal structure and use it to regularize the predictor of causal effects. The method is interesting and looks effective, although this assumption may not hold always true (e.g., in some domains, some causal influences may disappear, leading to extra conditional independence relations). Hope the authors will update the paper to address the concerns raised by the reviewers, especially to conduct a sensitivity analysis of the framework to misspecification of the causal structure and make the motivation for the used evaluation metrics clear, and also provide a more thorough review of related work.